# Dlg1 activates beta-catenin signaling to regulate retinal angiogenesis and the blood-retina and blood-brain barriers

**Chris Cho[1], Yanshu Wang[1,2], Philip M Smallwood[1,2], John Williams[1,2], Jeremy Nathans[1,2,3,4]***

[1]Department of Molecular Biology and Genetics, Johns Hopkins University School of Medicine, Baltimore, United States; [2]Howard Hughes Medical Institute, Johns Hopkins University School of Medicine, Baltimore, United States; [3]Department of Neuroscience, Johns Hopkins University School of Medicine, Baltimore, United States; [4]Department of Ophthalmology, Johns Hopkins University School of Medicine, Baltimore, United States

**Abstract** Beta-catenin (i.e., canonical Wnt) signaling controls CNS angiogenesis and the blood-brain and blood-retina barriers. To explore the role of the Discs large/membrane-associated guanylate kinase (Dlg/MAGUK) family of scaffolding proteins in beta-catenin signaling, we studied vascular endothelial cell (EC)-specific knockout of Dlg1/SAP97. EC-specific loss of Dlg1 produces a retinal vascular phenotype that closely matches the phenotype associated with reduced beta-catenin signaling, synergizes with genetically-directed reductions in beta-catenin signaling components, and can be rescued by stabilizing beta-catenin in ECs. In reporter cells with CRISPR/Cas9-mediated inactivation of Dlg1, transfection of Dlg1 enhances beta-catenin signaling ~4 fold. Surprisingly, Frizzled4, which contains a C-terminal PDZ-binding motif that can bind to Dlg1 PDZ domains, appears to function independently of Dlg1 in vivo. These data expand the repertoire of Dlg/MAGUK family functions to include a role in beta-catenin signaling, and they suggest that proteins other than Frizzled receptors interact with Dlg1 to enhance beta-catenin signaling.
DOI: https://doi.org/10.7554/eLife.45542.001

*For correspondence:
jnathans@jhmi.edu

## Introduction

The mammalian retina depends on two distinct vascular networks: the choroidal and intra-retinal circulations, which supply the outer third and inner two-thirds of the retina, respectively (*Sun and Smith, 2018*). The intra-retinal vessels originate from the optic nerve and, in the first stage of their development, proliferate radially to cover the vitreal surface of the retina. Vessels from this superficial plexus then grow into the inner retina and form two additional tiers of capillary beds – the deep and intermediate plexuses.

Mature vascular endothelial cells (ECs) of the central nervous system (CNS), including the retina, form a specialized diffusion barrier between blood and tissue known as the blood-brain barrier (BBB), and its retinal analog, the blood-retina barrier (BRB) (*Daneman and Prat, 2015*). The specialized features of CNS ECs that account for the barrier include: (i) the presence of tight junctions; (ii) the absence of fenestrations; (iii) suppression of transcytosis; and (iv) expression of small molecule transporters and extrusion pumps. The BBB and BRB allow for the selective transport of small molecules across the endothelium, thereby protecting the surrounding neurons and glia from toxins and other bioactive molecules.

The growth and differentiation programs of CNS ECs are controlled by multiple signaling systems, including beta-catenin (canonical Wnt) signaling (*Xu et al., 2004*; *Stenman et al., 2008*;

*Liebner et al., 2008*; *Daneman et al., 2009*). In the embryonic brain and spinal cord, neuroepithelium-derived Wnt7a and Wnt7b are the principal beta-catenin signaling ligands required for angiogenesis and BBB development (*Stenman et al., 2008*; *Daneman et al., 2009*). In the adult brain, Wnt7a and Wnt7b are produced predominantly by glia (*Wang et al., 2018a*). These ligands act on ECs through a Frizzled (Fz) receptor, Lrp5/6 co-receptor, and two Wnt7a/b-specific cell-surface co-activators, Reck and Gpr124 (*Zhou and Nathans, 2014*; *Posokhova et al., 2015*; *Vanhollebeke et al., 2015*; *Cho et al., 2017*; *Eubelen et al., 2018*; *Vallon et al., 2018*). In the retina, Müller glia-derived Norrin is the predominant beta-catenin signaling ligand responsible for angiogenesis and BRB development (*Ye et al., 2009*; *Wang et al., 2018a*). Norrin stimulates beta-catenin signaling in retinal ECs by acting on Fz4, Lrp5, and the Norrin-specific cell-surface co-activator Tspan12 (*Xu et al., 2004*; *Junge et al., 2009*). In humans, loss-of-function mutations in each of these Norrin signaling components causes retinal hypovascularization, a phenotype that is also observed in the corresponding gene knockout mouse lines (*Wang et al., 2018b*).

On the cytosolic side of the membrane, beta-catenin signaling involves ligand-dependent receptor/co-receptor regulation of the beta-catenin destruction complex (*MacDonald et al., 2009*). The dynamic association, dissociation, internalization, and recycling of receptors, co-receptors, and associated proteins during beta-catenin signaling suggest a potential role for scaffolding proteins in orchestrating this process. Earlier studies showed that Discs large homologue 1 [Dlg1; also called Dlgh1 or synapse-associated protein 97 (SAP97)], an intracellular scaffolding protein with PDZ and guanylate kinase domains, can bind to a PDZ-binding motif present at the C-terminus of several Frizzleds, including Fz4 (*Hering and Sheng, 2002*). These data suggest that Dlg1 or other Dlg family members could play a role in beta-catenin signaling, presumably via an effect on trafficking or localization of Frizzled receptors.

While Dlg family members have been widely studied in the context of cell polarity, synaptic organization, and protein trafficking (*Yamanaka and Ohno, 2008*; *Walch, 2013*; *Won et al., 2017*), their potential connection to beta-catenin signaling is largely unexplored, and, to our knowledge, is limited to only two reports, one in Drosophila that implies a stimulatory role for Dlg via stabilization of Disheveled (*Liu et al., 2016*) and the second in mammalian cell culture that implies an inhibitory role for Dlg1 via enhanced GSK3β phosphorylation of beta-catenin followed by ubiquitination and proteolysis (*Boccitto et al., 2016*). The mechanism of stimulation described for Drosophila Dlg involves a Disheveled-binding motif that is conserved in mammalian Dlg2 but is absent from the four other mammalian Dlg family members (*Liu et al., 2016*). In P7 mouse brain ECs, RNAseq shows that *Dlg1* transcripts (18 FPKM) are ~7 fold more abundant than transcripts coding for the next most abundant family member, *Dlg5* (2.5 FPKM); transcripts for other *Dlg* family members are present at even lower levels [*Dlg2* (0 FPKM), *Dlg3* (1.5 FPKM), and *Dlg4* (1.5 FPKM)] (*Zhang et al., 2014*).

In the present study, we have identified a role for endothelial Dlg1 in retinal angiogenesis and BBB/BRB development as determined by the phenotype of mice with EC-specific knockout of *Dlg1*. We provide evidence for a genetic interaction between *Dlg1* and *Fz4*, *Tspan12*, and *Ndp* (the gene coding for Norrin), which suggests that Dlg1 stimulates beta-catenin signaling in CNS ECs. This idea is supported by the finding that Dlg1 enhances beta-catenin signaling in cultured cells and by rescue of the *Dlg1* EC-specific knockout phenotype upon constitutive beta-catenin stabilization in vivo. Intriguingly, although the first two PDZ domains of Dlg1 bind in vitro to a consensus PDZ-binding motif at the Fz4 C-terminus, genetic epistasis analyses of the EC-specific *Dlg1* knockout and a CRISPR/Cas9-generated mutation of the Fz4 PDZ-binding motif argues that these two proteins function independently in the context of beta-catenin signaling.

## Results

### Vascular endothelial depletion of Dlg1 impairs retinal angiogenesis

Ubiquitous loss of *Dlg1* leads to neonatal lethality (*Caruana and Bernstein, 2001*; *Mahoney et al., 2006*; *Rivera et al., 2013*; *Iizuka-Kogo et al., 2007*; *Iizuka-Kogo et al., 2015*). Therefore, to study the role of *Dlg1* in CNS vascular development, we utilized a conditional allele (*Dlg1^{fl}*; *Zhou et al., 2008*) and *Tie2-Cre*, which recombines at high efficiency in ECs starting at mid-gestation. We began by analyzing *Dlg1^{fl/Δ};Tie2-Cre* retinas at three postnatal (P) time points: P7, P14, and P18 (*Figure 1*). At these ages and throughout adulthood, *Dlg1^{fl/Δ};Tie2-Cre* mice were indistinguishable in overall

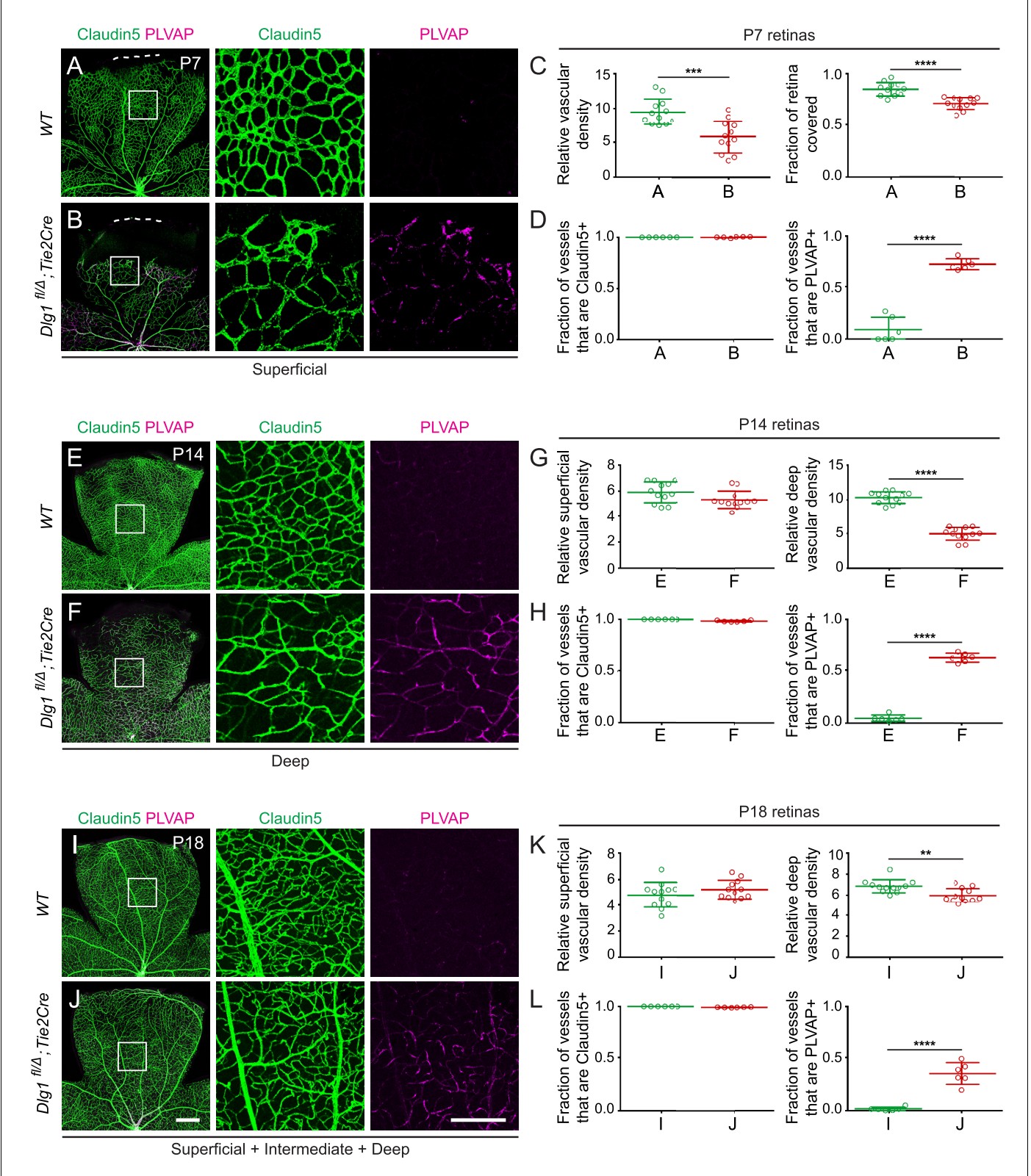

**Figure 1.** Dlg1 promotes angiogenesis in the mammalian retina. (**A–B**) The superficial vascular plexus of P7 retinas from the indicated genotypes (column 1), with boxed regions shown at higher magnification (columns 2 and 3). Dashed white lines indicate the edge of the retina. (**C**) Quantification of vascular density (left) and vascular coverage (right) in P7 retinas, for the genotypes in (**A**) and (**B**). Quantification methodology in this and other figures is described in the Materials and methods section. In this and all other figures, bars represent mean ± SD. Statistical significance, determined by the

*Figure 1 continued on next page*

*Figure 1 continued*

unpaired t-test, is represented by * (p<0.05), ** (p<0.01), *** (p<0.001), and **** (p<0.0001). (D) Quantification of the fraction of vessel length that is Claudin5+ (left) and PLVAP+ (right) in P7 retinas, for the genotypes in (A) and (B). (E–F) The deep vascular plexus of P14 retinas (column 1) with boxed regions shown at higher magnification (columns 2 and 3). (G) Quantification of vascular density in the superficial plexus (left) and deep plexus (right) in P14 retinas for the genotypes in (E) and (F). (H) As in (D), except with P14 retinas. (I–J) Maximum projection of superficial, intermediate, and deep vascular plexuses of P18 retinas (column 1) with boxed regions shown at higher magnification (columns 2 and 3). (K) As in (G), except with P18 retinas. (L) As in (D), except with P18 retinas. Scale bar for column 1, 400 µm. Scale bar for columns 2 and 3, 200 µm.

DOI: https://doi.org/10.7554/eLife.45542.002

The following figure supplement is available for figure 1:

**Figure supplement 1.** Quantification of Claudin5 and PLVAP accumulation in retinal ECs.

DOI: https://doi.org/10.7554/eLife.45542.003

appearance and health from their WT littermates. At P7, *Dlg1^{fl/Δ}*;*Tie2-Cre* retinas displayed reduced density and retarded radial growth of the superficial vascular plexus relative to WT controls (*Figure 1A and B*; quantified in *Figure 1C*). At P14, the density of the superficial vascular plexus in *Dlg1^{fl/Δ}*;*Tie2-Cre* retinas appeared normal, while the deep vascular plexus showed reduced density (*Figure 1E and F*; quantified in 1G). By P18, the density of the deep vascular plexus in *Dlg1^{fl/Δ}*;*Tie2-Cre* retinas had nearly caught up to the WT control (*Figure 1I and J*; quantified in K).

At all three developmental stages, a subset of ECs in *Dlg1^{fl/Δ}*;*Tie2-Cre* retinas expressed the fenestral diaphragm protein plasmalemma vesicle-associated protein (PLVAP), which was undetectable in WT controls (*Figure 1A and B,E and F,I and J*; quantified in D, H, and L, respectively). Retarded angiogenesis and induction of PLVAP are hallmarks of reduced beta-catenin signaling in ECs, as seen with loss of *Fz4*, *Tspan12*, *Lrp5*, or *Ndp*, and they correlate with the degree of reduction in beta-catenin signaling (*Xia et al., 2008*; *Junge et al., 2009*; *Wang et al., 2012*; *Rattner et al., 2014*). The retarded vascular growth observed in *Dlg1^{fl/Δ}*;*Tie2-Cre* P7 retinas with a vascular density that nearly approximates WT retinas by P18 is consistent with a modest reduction in beta-catenin signaling (*Figure 1A and B,I and J*; quantified in C and K, respectively). ECs in *Dlg1^{fl/Δ}*; *Tie2-Cre* retinas also showed a modest reduction in the abundance of the tight junction marker Claudin5 and a subset of ECs co-expressed Claudin5 and PLVAP, but there was no conversion of ECs from Claudin5+/PLVAP- to Claudin5-/PLVAP+, as seen in retinas with a severe decrement in EC beta-catenin signaling (*Wang et al., 2012*; *Wang et al., 2018a*). Representative images and tracings of Claudin5+ and PLVAP+ retinal vessels – the method used for quantifying EC marker expression – are shown in *Figure 1—figure supplement 1*. In sum, EC-specific loss of Dlg1 produces a mild retinal vascular phenotype that is consistent with a modest reduction in beta-catenin signaling.

## Evidence for a genetic interaction between *Dlg1* and *Fz4*

In light of the phenotypic evidence suggesting that Dlg1 might promote beta-catenin signaling in ECs, we asked whether *Dlg1* interacts genetically with *Fz4*. Since heterozygous loss of *Fz4* confers a near-normal vascular phenotype (*Zhou et al., 2014*), we reasoned that if Dlg1 promotes beta-catenin signaling in ECs, then the combination of EC-specific loss of *Dlg1* and heterozygous loss of *Fz4* might produce a retinal vascular phenotype more severe than either alone. *Figure 2* shows that this is the case: at P18, *Dlg1^{fl/Δ}*;*Fz4^{+/Δ}*;*Tie2-Cre* retinas display a dramatic reduction in vascular density specifically in the deep vascular plexus and widespread expression of PLVAP in retinal ECs. By comparison, *Dlg1^{fl/Δ}*;*Tie2-Cre* and *Dlg1^{fl/+}*;*Fz4^{+/Δ}*;*Tie2-Cre* retinas are nearly indistinguishable from WT (*Figures 1J* and *2A*; quantified in *Figure 2C–E*). BRB integrity was assessed with the membrane-impermeable sulfo-N-hydroxysuccinimide-long chain-biotin (sulfo-NHS-biotin) tracer, introduced into the intravascular space by intraperitoneal (IP) injection. In *Dlg1^{fl/Δ}*;*Fz4^{+/Δ}*;*Tie2-Cre* retinas, the BRB was minimally impaired, as judged by the low level of sulfo-NHS-biotin leakage (*Figure 2B*). These data provide evidence for a genetic interaction between *Dlg1* and *Fz4* in controlling retinal angiogenesis and BRB-related gene expression. [In this and all subsequent genetic interaction experiments, we emphasize that 'interaction' does not connote a 'physical interaction', but rather it means that the phenotype of two mutations in combination differs from the simple addition or superposition of the individual mutant phenotypes. Additionally, in this and all subsequent experiments, phenotypically WT but not genotypically *WT* littermates were used as controls because, depending on

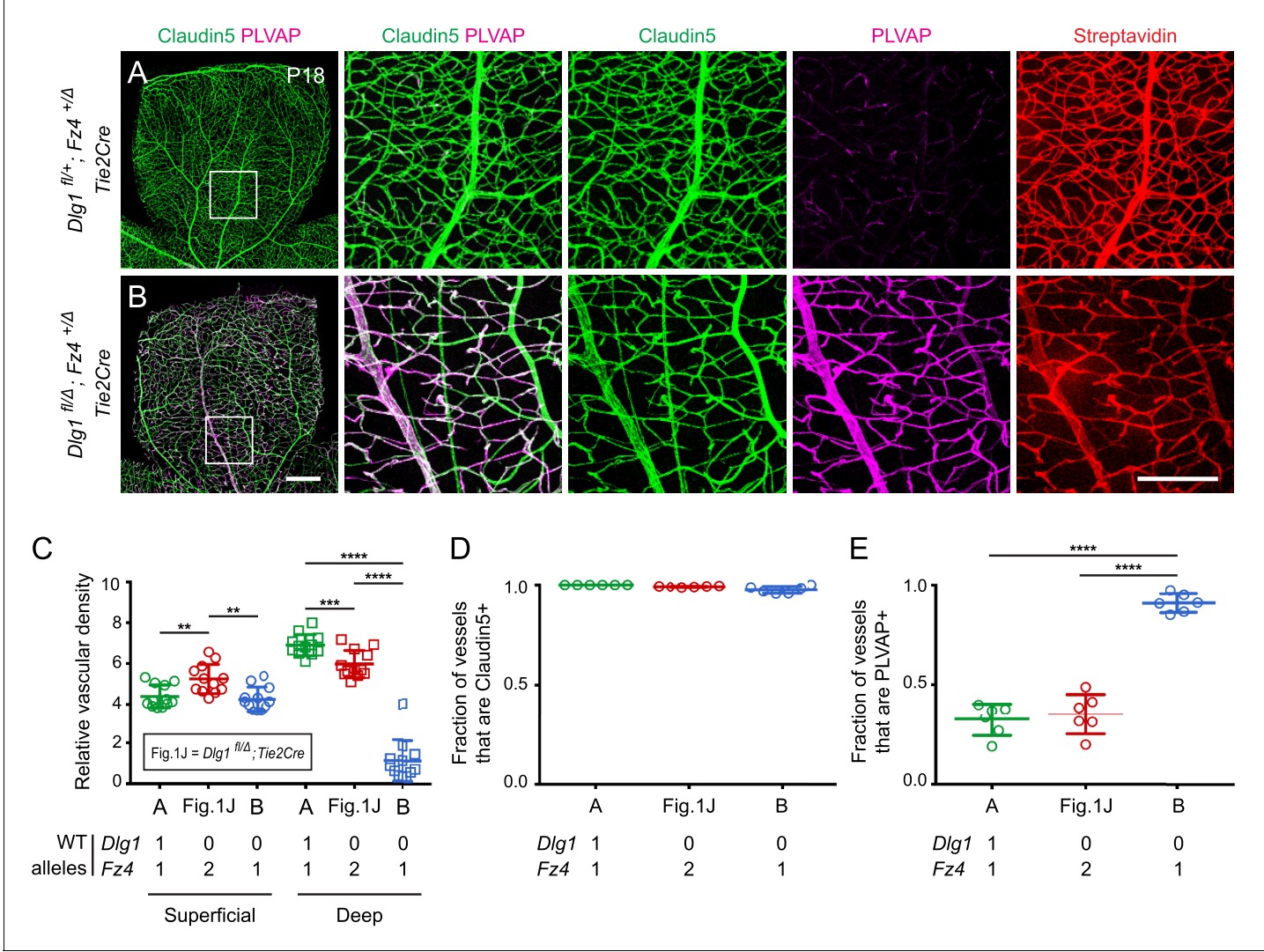

**Figure 2.** *Dlg1* genetically interacts with *Fz4* to regulate retinal angiogenesis and the BRB. (**A–B**) Maximum projection of the superficial, intermediate, and deep vascular plexuses of P18 retinas from the indicated genotypes (column 1), with boxed regions displayed at higher magnification in columns 2–5. Mice were injected IP with 2–3 mg of sulfo-NHS-biotin 1–2 hr before sacrifice. Scale bar for column 1, 400 μm. Scale bar for columns 2–5, 200 μm. (**C–E**) Quantification of vascular density in the superficial and deep plexuses (**C**), the fraction of vessels that immunostain for Claudin5 (**D**), and the fraction of vessels that immunostain for PLVAP (**E**), for the genotypes shown in (**A**), (**B**), and *Figure 1J*. In this Figure and *Figures 3* and *4*, the numbers of WT alleles for the genes of interest are listed below each set of data points.

DOI: https://doi.org/10.7554/eLife.45542.004

the cross, the genetic crosses used to generate the experimental mice either rarely or never yielded genotypically *WT* littermates.]

## *Dlg1* genetically interacts with *Tspan12* to control retinal angiogenesis and the BRB

Given the evidence that *Dlg1* and *Fz4* interact genetically, we next asked whether other components of the Norrin-Fz4 signaling pathway also interact genetically with *Dlg1*. Loss of Tspan12, a receptor co-factor for Norrin-Fz4 signaling, leads to a retinal vascular phenotype that closely resembles, but is less severe than, the phenotype seen with loss of Norrin or Fz4 (*Junge et al., 2009*). We therefore asked whether (i) the combined loss of one or more alleles of *Dlg1* and *Tspan12* produces more severe retinal vascular phenotypes as more WT alleles are lost and (ii) whether the phenotype

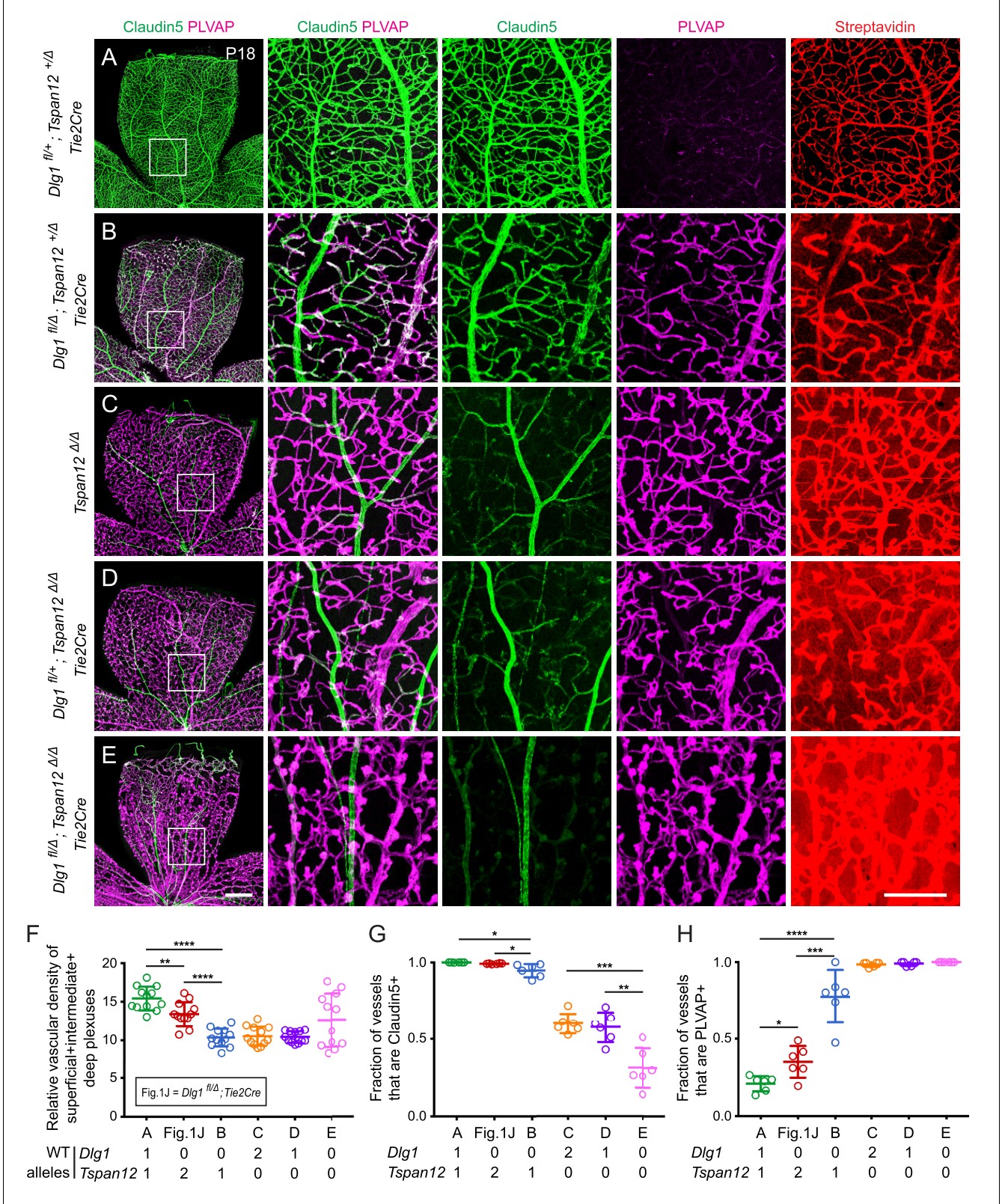

**Figure 3.** *Dlg1* genetically interacts with *Tspan12* in the developing retinal vasculature. (A–E) Maximum projection of the superficial, intermediate, and deep vascular plexuses of P18 retinas from the indicated genotypes (column 1), with boxed regions displayed at higher magnification in columns 2–5. Mice were injected IP with 2–3 mg of sulfo-NHS-biotin 1–2 hr before sacrifice. Scale bar for column 1, 400 μm. Scale bar for columns 2–5, 200 μm. (F–H)

*Figure 3 continued*

Quantification of vascular density (**F**), the fraction of vessels that immunostain for Claudin5 (**G**), and the fraction of vessels that immunostain for PLVAP (**H**), for the genotypes shown in (**A–E**) and *Figure 1J*.

DOI: https://doi.org/10.7554/eLife.45542.005

associated with the combined loss of both genes is more severe than the phenotypes associated with the loss of either gene alone (*Figure 3A–E*; quantification in F-H). Analyses were conducted at P18.

$Dlg1^{fl/+};Tspan12^{+/\Delta};Tie2$-Cre retinas (i.e. with one WT allele for each gene; *Figure 3A*) displayed essentially normal vascular architecture, barely detectable PLVAP expression, and no detectable sulfo-NHS-biotin leakage. $Dlg1^{fl/\Delta};Tspan12^{+/\Delta};Tie2$-Cre retinas (i.e. no WT $Dlg1$ alleles and one WT $Tspan12$ allele; *Figure 3B*) displayed a more severe vascular phenotype compared to $Dlg1^{fl/\Delta};Tie2$-Cre and $Dlg1^{fl/+};Tspan12^{+/\Delta};Tie2$-Cre (compare to *Figures 1J* and *3A*, respectively). Specifically, $Dlg1^{fl/\Delta};Tspan12^{+/\Delta};Tie2$-Cre retinas showed a greater reduction in vascular density, reduced Claudin5, increased PLVAP, and minimal sulfo-NHS-biotin leakage. $Tspan12^{\Delta/\Delta}$ retinas (*Figure 3C*) and $Dlg1^{fl/+};Tspan12^{\Delta/\Delta};Tie2$-Cre retinas (i.e. a single $Dlg1$ allele and no $Tspan12$ alleles; *Figure 3D*) showed reduced density of the vasculature, further reductions in Claudin5, and widespread accumulation of PLVAP. While both genotypes showed a moderate level of sulfo-NHS-biotin leakage, leakage appears to be incrementally higher in $Dlg1^{fl/+};Tspan12^{\Delta/\Delta};Tie2$-Cre retinas.

The complete loss of both $Dlg1$ and $Tspan12$ ($Dlg1^{fl/\Delta};Tspan12^{\Delta/\Delta};Tie2$-Cre; *Figure 3E*) produced the most severe retinal vascular phenotype, comparable to the phenotypes produced by loss of $Ndp$ or $Fz4$, with nearly complete conversion of veins and capillaries from Claudin5+/PLVAP- to Claudin5-/PLVAP+ and severe sulfo-NHS-biotin leakage. As described previously, arteries do not convert to a PLVAP+ state with loss of beta-catenin signaling (*Wang et al., 2012*). The variable increase in vascular density in $Dlg1^{fl/\Delta};Tspan12^{\Delta/\Delta};Tie2$-Cre retinas (quantified in *Figure 3F*) reflects the hypertrophy of vascular glomeruloids, a secondary response to vascular endothelial growth factor (VEGF) secreted by the severely hypoxic retina and a hallmark of $Ndp$ and $Fz4$ mutant retinas (*Ye et al., 2009*; *Rattner et al., 2014*). Together, these data provide evidence for a strong genetic interaction between $Dlg1$ and $Tspan12$, further supporting a role for Dlg1 in promoting retinal angiogenesis and BRB development via the Norrin/Fz4/Tspan12 pathway.

## *Dlg1* functions redundantly with *Ndp* and *Tspan12* to regulate the blood-brain barrier

We next asked whether endothelial Dlg1 plays a role in CNS vascular biology beyond the retina. As noted in the Introduction, beta-catenin signaling in brain ECs depends on the partially redundant Wnt7/Fz/Gpr124/Reck and Norrin/Fz4/Tspan12 signaling systems and is required for the development and maintenance of the BBB (*Wang et al., 2012*; *Zhou et al., 2014*; *Zhou and Nathans, 2014*; *Cho et al., 2017*; *Wang et al., 2018a*). Genetic loss of these signaling components compromises the BBB in a graded and region-specific fashion, with progressive decrements in beta-catenin signaling producing progressively more severe defects in BBB integrity (*Wang et al., 2012*; *Wang et al., 2018a*; *Zhou et al., 2014*; *Zhou and Nathans, 2014*; *Cho et al., 2017*). For example, $Ndp$ null mice have mild BBB defects, primarily in the cerebellum and olfactory bulb, while $Tspan12$ null mice have no BBB defects, consistent with the more modest effect of Tspan12 on Norrin/Fz4/Tspan12 signaling (*Junge et al., 2009*; *Wang et al., 2018a*). In the cerebellum, signaling by the Wnt7/Fz/Gpr124/Reck system is apparently sufficient to compensate for the loss of $Tspan12$, but it does not fully compensate for the loss of $Ndp$ (*Zhou et al., 2014*; *Wang et al., 2018a*).

In the present experiments, we have examined brains at P18 and we have asked whether loss of $Tspan12$ or $Ndp$ combined with endothelial-specific loss of one or both $Dlg1$ alleles elicits or exacerbates a BBB defect (*Figure 4A–D*; quantification in *Figure 4E and F*). Since previous work has shown that the vasculature in the cerebellum is especially sensitive to the level of endothelial beta-catenin signaling (*Zhou et al., 2014*; *Wang et al., 2018a*), we have visualized cerebellar sulfo-NHS-biotin leakage and quantified cerebellar EC expression of PLVAP and the glucose transporter GLUT1, a marker of BBB-competent ECs.

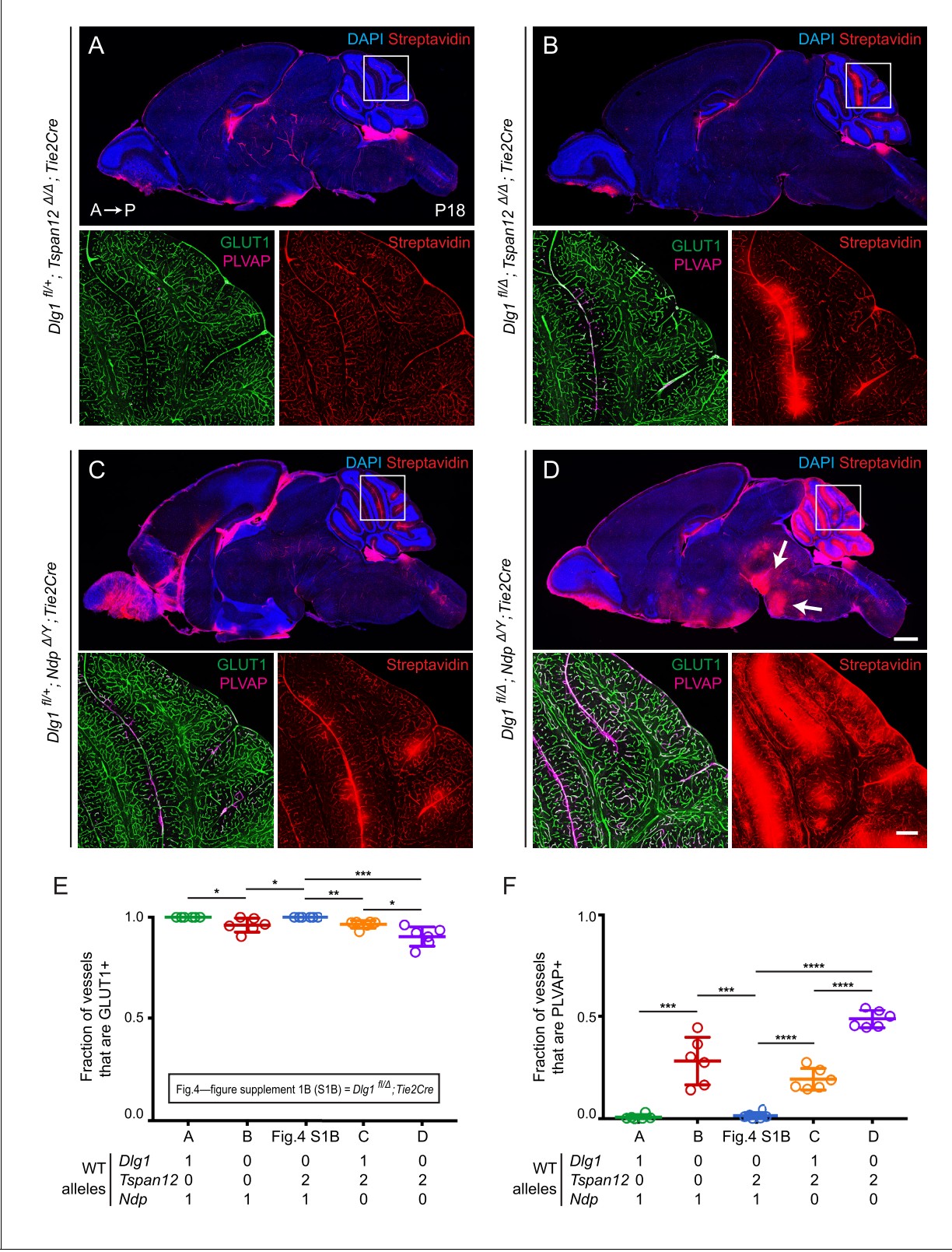

**Figure 4.** *Dlg1* genetically interacts with *Tspan12* and *Ndp* to regulate BBB integrity. (**A–D**) Sagittal sections of P18 brains from the indicated genotypes, with color channels for the indicated histochemical stains and/or immunostains indicated in each image. In each set, the boxed region in the cerebellum in the upper image is shown at higher magnification in the lower two images. Mice were injected IP with 2–3 mg of sulfo-NHS-biotin 1–2 hr before sacrifice. (*Ndp* is an X-linked gene, and Norrin null males are designated *Ndp*<sup>Δ/Y</sup>.) Arrows in (**D**) point to biotin leakage in the hindbrain. A,

*Figure 4 continued on next page*

*Figure 4 continued*

anterior; P, posterior. Scale bar (low magnification images), 1 mm. Scale bar (high magnification images), 200 μm. (**E,F**) Quantification of the fraction of vessels that immunostain for GLUT1 (**E**) and PLVAP (**F**), for each genotype shown in (**A–D**) and *Dlg1 $^{fl/\Delta}$; Tie2-Cre* (***Figure 4—figure supplement 1B***).
DOI: https://doi.org/10.7554/eLife.45542.006

The following figure supplement is available for figure 4:

**Figure supplement 1.** Loss of *Dlg1* in endothelial cells has no effect on vascular leakage or vascular markers GLUT1 and PLVAP.
DOI: https://doi.org/10.7554/eLife.45542.007

Halving the dose of endothelial *Dlg1* (*Dlg1$^{fl/+}$;Tie2-Cre*), eliminating endothelial *Dlg1* (*Dlg1$^{fl/\Delta}$; Tie2-Cre*), or halving the dosage of endothelial *Dlg1* on the background of a *Tspan12* knockout (*Dlg1$^{fl/+}$;Tspan12$^{\Delta/\Delta}$;Tie2-Cre*) produced no loss of GLUT1 expression, no increase in PLVAP expression, and no BBB breakdown as assessed by sulfo-NHS-biotin leakage (***Figure 4—figure supplement 1***; ***Figure 4A***). However, a complete loss of endothelial *Dlg1* on the background of a *Tspan12* knockout (*Dlg1$^{fl/\Delta}$;Tspan12$^{\Delta/\Delta}$;Tie2-Cre*) partially compromises the cerebellar BBB, as seen by a small decrease in the fraction of GLUT1+ ECs, an increase in the fraction of PLVAP+ ECs, and localized sulfo-NHS-biotin leakage (***Figure 4B***). Similarly, a comparison of heterozygous loss of *Dlg1* on the background of a *Ndp* knockout (*Dlg1$^{fl/+}$;Ndp$^{\Delta/Y}$;Tie2-Cre*) vs. homozygous loss of *Dlg1* on the background of a *Ndp* knockout (*Dlg1$^{fl/\Delta}$;Ndp$^{\Delta/Y}$;Tie2-Cre*) shows a dramatic worsening of BBB breakdown in both the cerebellum and hindbrain with loss of both *Dlg1* alleles (compare ***Figure 4C and D***). The severe *Dlg1$^{fl/\Delta}$;Ndp$^{\Delta/Y}$;Tie2-Cre* phenotype closely resembles the phenotypes observed when both Norrin/Fz4 and Wnt7/Fz/Gpr124/Reck signaling are compromised (***Zhou et al., 2014***; ***Wang et al., 2018a***).

We note that the greater BBB breakdown in *Dlg1$^{fl/\Delta}$;Ndp$^{\Delta/Y}$;Tie2-Cre* brains compared to *Dlg1$^{fl/+}$;Ndp$^{\Delta/Y}$;Tie2-Cre* brains indicates that Dlg1 function is not limited to the Norrin/Fz4/Tspan12 signaling pathway, which is eliminated with loss of *Ndp*. The simplest explanation for the genetic interactions shown in ***Figures 2–4*** is that Dlg1 normally promotes both Norrin/Fz4/Tspan12 signaling and Wnt7/Fz/Gpr124/Reck signaling.

## Beta-catenin stabilization rescues the angiogenesis and BRB defects caused by endothelial-specific *Dlg1* deletion

To directly test the idea that Dlg1 promotes beta-catenin signaling in CNS ECs, we have asked whether the phenotypes caused by loss of Dlg1 can be rescued by artificially activating beta-catenin signaling in a manner that likely bypasses Dlg1 function. Although the site of Dlg1 function has not been defined by the genetic interaction experiments described above, by analogy with Dlg1 function in other contexts, it is a reasonable guess that Dlg1 organizes membrane-associated proteins in ECs. In the context of beta-catenin signaling, Dlg1 could interact directly or indirectly with receptors, co-receptors, the beta-catenin destruction complex, and/or regulators of the destruction complex. With this model in mind, we have asked whether the *Dlg1* null phenotype can be rescued by EC-specific expression of a stabilized derivative of beta-catenin that leads to constitutive signaling independent of ligand/receptor and destruction complex function. More specifically, this experiment involves Cre-mediated excision of *loxP*–flanked exon three from one copy of the beta-catenin gene (*Ctnnb1*; this conditional allele is referred to as *Ctnnb1$^{flex3}$*; ***Harada et al., 1999***) using an EC-specific *Pdgfb-CreER* transgene. Exon three encompasses the sites of phosphorylation that lead to beta-catenin degradation, and its excision conveniently maintains the *Ctnnb1* open reading frame and does not appear to adversely affect beta-catenin function. For this experiment, recombination was induced with CreER and 4-hydroxytamoxifen (4HT) treatment in the immediate postnatal period because constitutive EC-specific activation of beta-catenin signaling, as produced by *Ctnnb1$^{flex3/+}$; Tie2-Cre*, is lethal during embryonic development.

We analyzed retinas at P7, an age when constitutive loss of endothelial *Dlg1* (*Dlg1$^{fl/\Delta}$;Tie2Cre*) produces a substantially reduced vascular density compared to WT controls (***Figure 1B***; quantified in ***Figure 1C***). At this age, the *Dlg1$^{fl/\Delta}$;Tie2Cre* retinal vasculature also shows increased PLVAP accumulation and sulfo-NHS-biotin leakage at the leading edge of the plexus (***Figure 1B***; ***Figure 5—figure supplement 1***; quantified in ***Figure 1D***). Consistent with the *Dlg1$^{fl/\Delta}$;Tie2Cre* phenotype, *Dlg1$^{fl/\Delta}$;Pdgfb-CreER* mice that were treated with 4HT at P0 and P1 showed reduced vascular density,

increased PLVAP levels, and extensive sulfo-NHS-biotin leakage when compared to 4HT-treated control mice ($Dlg1^{fl/\Delta}$) (**Figure 5A–C**; quantified in **Figures 5E, G and H**; two examples of $Dlg1^{fl/\Delta}$; *Pdgfb-CreER* retinas are shown to show the consistency of the phenotype). We note that the vascular coverage is more similar in a comparison between $Dlg1^{fl/\Delta}$;*Pdgfb-CreER* vs. $Dlg1^{fl/\Delta}$ control retinas (**Figure 5A–C**; quantified in **Figure 5F**) compared to $Dlg1^{fl/\Delta}$;*Tie2Cre* vs. *WT* controls (**Figure 1B**; quantified in **Figure 1C**). This difference could arise from the use of an inducible *Pdgfb-CreER*, which may allow for normal radial growth prior to the time of recombination at P0-P1.

Strikingly, the retinal vascular defects in $Dlg1^{fl/\Delta}$;*Pdgfb-CreER* mice were almost completely reversed upon EC-specific stabilization of beta-catenin ($Dlg1^{fl/\Delta}$; $Ctnnb1^{flex3/+}$;*Pdgfb-CreER*), as seen in **Figure 5D** and quantified in **Figure 5E–H**. These experiments lend strong support to a model in which Dlg1 in ECs promotes retinal angiogenesis and the development of the BRB by stimulating beta-catenin signaling.

## Dlg1 stimulates beta-catenin signaling in a reporter cell line

To further assess the role of Dlg1 in beta-catenin signaling, we began by asking whether over-production of Dlg1 enhances signaling in a Super TOP Flash (STF) luciferase reporter cell line in the presence of Norrin and Fz4, which together produce a 20-50x induction of luciferase over background (**Xu et al., 2004**). **Figure 6A** shows a bell-shaped curve of signal strength vs. N-terminal Myc-epitope-tagged Dlg1 (Myc-Dlg1) plasmid concentration, with an optimal dose at 2.5 ng/well and a maximum stimulation of ~2 fold. One possible explanation for this modest degree of stimulation is that endogenous Dlg1 and/or other Dlg family proteins provide Dlg activity in STF cells. Consistent with this idea, RNAseq shows that STF cells have substantial levels of transcripts coding for Dlg1 (14 FPKM), Dlg3 (20 FPKM), and the divergent family member Dlg5 (7.4 FPKM), with lower levels of transcripts coding for Dlg2 (0.1 FPKM) and Dlg4 (0.8 FPKM).

To eliminate any effects on beta-catenin signaling due to endogenous Dlg1, we generated two *Dlg1* knockout (KO) cell lines (Clone #12 and Clone #15) from the starting STF cell line using CRISPR/Cas9 (**Figure 6—figure supplement 1**). Transfection of both knockout cell lines with Norrin and Fz4, together with Myc-Dlg1, showed (i) a lower baseline level of beta-catenin signaling without transfected Dlg1 and (ii) a dose response curve similar to the one obtained with the parental STF line, except that the optimal Myc-Dlg1 plasmid concentration was increased ~1.5 fold and the maximal fold stimulation increased from ~2 fold to ~4 fold (**Figure 6B and C**). These data are consistent with the hypothesis that endogenous Dlg1 promotes beta-catenin signaling. We interpret the reduced signal strength at higher Myc-Dlg1 plasmid concentrations (10 and 20 ng/well) as likely due to a non-optimal stoichiometry between over-produced Myc-Dlg1 and less abundant endogenous proteins. In the experiments described below, we used *Dlg1* KO STF clone #15.

To determine whether Myc-Dlg1 could enhance beta-catenin signaling that was activated by Wnt ligands, we transfected *Dlg1* KO STF cells with Fz4 and each of the 19 Wnts or Norrin, with or without Myc-Dlg1. We observed a 3- to 4-fold increase in luciferase activity for the majority of Wnt ligands upon co-transfection with Myc-Dlg1 (**Figure 6D**). Interestingly, co-expressed Myc-Dlg1 had little effect on signaling by Wnt7a and Wnt7b (**Figure 6D**). In vivo, Gpr124 and Reck function as specific co-factors for Wnt7a and Wnt7b signaling. Although the C-terminus of Gpr124 contains a conserved PDZ-binding domain that has been reported to directly interact with Dlg1 (**Yamamoto et al., 2004**; **Posokhova et al., 2015**), Myc-Dlg1 co-transfection of *Dlg1* KO STF cells transfected with Wnt7a, Gpr124, and Reck increased beta-catenin signaling by only ~1.5 fold at the optimal Myc-Dlg1 plasmid concentration of 2.5 ng/well (**Figure 6E**). Thus, in the context of transfected STF cells, Dlg1 plays only a modest role in Wnt7a/Fz/Gpr124/Reck signaling.

We next explored whether other mammalian Dlg family members could activate beta-catenin signaling in STF cells. Dlg constructs with an N-terminal Myc tag (Dlg1-Dlg4) or V5 tag (Dlg5) showed protein accumulation at the expected molecular weights in transfected HEK/293T cells (**Figure 6F**). In *Dlg1* KO STF cells transfected with Norrin, Fz4, and the individual Dlg family members, robust stimulation of beta-catenin signaling was produced by Myc-Dlg1, Myc-Dlg2, and Myc-Dlg4, with a bell-shaped (Dlg1) or monotonic (Dlg2 and Dlg4) plasmid concentration dependence (**Figure 6G**). In contrast, Myc-Dlg3 and V5-Dlg5 produced minimal stimulation (**Figure 6G**). These data imply that Dlg2 and Dlg4 are the Dlg family members most capable of substituting for Dlg1. We speculate that the difference in the shapes of the dose-response curves may be due to: (i) a requirement for higher concentrations of Myc-Dlg2 and Myc-Dlg4 to elicit a dominant-negative effect on beta-catenin

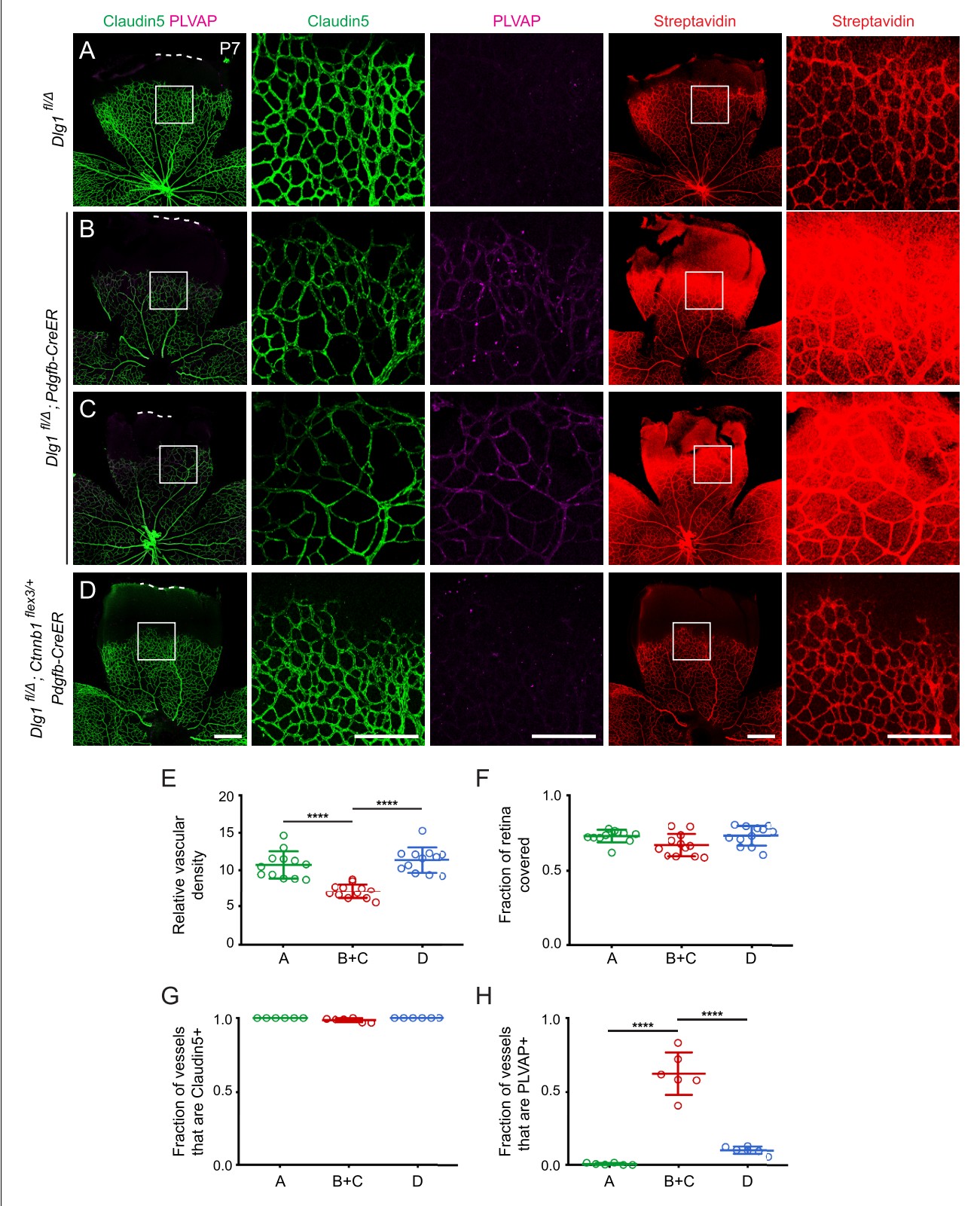

**Figure 5.** Stabilizing beta-catenin in endothelial cells corrects the retinal vascular defect caused by loss of Dlg1. (**A–D**) P7 retinas from the indicated genotypes, with the boxed region in column 1 shown at higher magnification in columns 2 and 3, and the boxed region in column 4 shown at higher magnification in column 5. Mice were injected IP with 2–3 mg of sulfo-NHS-biotin 1–2 hr before sacrifice. Dashed white lines indicate the edge of the retina. (**E–H**) Quantification of vascular density (**E**), vascular coverage (**F**), the fraction of vessels that immunostain for Claudin5 (**G**), and the fraction of

*Figure 5 continued on next page*

*Figure 5 continued*

vessels that immunostain for PLVAP (**H**), for the genotypes shown in (**A–D**). Scale bars for columns 1 and 4, 400 µm. Scale bars for columns 2, 3, and 5, 200 µm.

DOI: https://doi.org/10.7554/eLife.45542.008

The following figure supplement is available for figure 5:

**Figure supplement 1.** Loss of *Dlg1* in endothelial cells in P7 retinas: marker expression and biotin leakage.

DOI: https://doi.org/10.7554/eLife.45542.009

signaling; or (ii) the presence of one or more essential proteins that is/are sequestered by Myc-Dlg1 but not Myc-Dlg2 and Myc-Dlg4. Thus, in both CNS ECs and STF cells, where Dlg2 and Dlg4 transcripts are relatively rare, Dlg1 is likely to be the most important family member for beta-catenin signaling.

## Biochemical properties and cell culture activity of the Fz4 C-terminal PDZ-binding motif

As noted in the Introduction, the simplest and most attractive model for Dlg1 stimulation of beta-catenin signaling posits a direct interaction between one or more of the Dlg1 PDZ domains and the canonical PDZ-binding motif at the C-terminus of many Frizzled receptors, including Fz4. Using the yeast two-hybrid system, a previous study reported that a C-terminal peptide from Fz4, which ends with the consensus PDZ-binding motif ETVV [conforming to the type I C-terminal PDZ-binding motif consensus S/T-X-V/L (*Bezprozvanny and Maximov, 2001*), can interact directly with a fragment of Dlg1 comprising PDZ domains 1 and 2 (*Hering and Sheng, 2002*). To confirm and extend this observation, we first tested the binding of full-length Fz4 and full-length Dlg1, using a Rim epitope tag (*Illing et al., 1997*) between the signal sequence and the cysteine-rich domain (CRD) of Fz4 (Rim-Fz4) and Myc-Dlg1. In HEK/293T cells that were transiently co-transfected with Rim-Fz4 and Myc-Dlg1, co-immunoprecipitation (co-IP) with anti-Myc or anti-Rim antibodies showed that Myc-Dlg1 binds to the wild type (WT) version of Rim-Fz4 but not to a mutant version in which the four carboxyl-terminal residues were changed from ETVV to AAAA (*Figure 7A and B*). This experiment confirms that Dlg1 binding requires the Fz4 PDZ-binding motif.

To determine the domains of Myc-Dlg1 responsible for interacting with Rim-Fz4, we individually deleted each of the three tandem PDZ domains of Dlg1; the mutant proteins are referred to as ΔPDZ1, ΔPDZ2, and ΔPDZ3. Transient co-transfection of Rim-Fz4 with WT, ΔPDZ1, ΔPDZ2, or ΔPDZ3 versions of Myc-Dlg1 showed that: (i) with anti-Myc pull-down, co-IP efficiency is variably reduced for ΔPDZ1, substantially reduced for ΔPDZ2, and unaffected for ΔPDZ3, and (ii) with anti-Rim pull-down, co-IP efficiency is substantially reduced for both ΔPDZ1 and ΔPDZ2, and unaffected for ΔPDZ3 (*Figure 7C*). To further interrogate the Dlg1-Fz4 interaction, we assessed the binding of WT Dlg1 and its single and multiple PDZ deletion derivatives to an N-terminal biotinylated peptide consisting of the 12 carboxyl-terminal amino acids of Fz4 (*Figure 7D*). This experiment shows that, compared to WT Dlg1, ΔPDZ1 and ΔPDZ2 had greatly reduced peptide binding and ΔPDZ3 had modestly reduced peptide binding. Using the same assay, ΔPDZ2+3 (i.e. simultaneous deletion of domains 2 and 3) had greatly reduced binding, and ΔPDZ1+2 and ΔPDZ1+2+3 had little or no binding (*Figure 7E*). Finally, we purified (from *E. coli*) recombinant maltose binding protein (MBP) fusions with PDZ1, PDZ2, or PDZ1+2 and assessed their binding to (i) an N-terminal biotinylated peptide consisting of the 12 carboxyl-terminal amino acids of Fz4 and (ii) an analogous N-terminal biotinylated peptide consisting of the 11 carboxyl-terminal amino acids of bovine rhodopsin (*Figure 7F*). This experiment showed efficient and specific peptide capture of MBP-PDZ1+2, but not of MBP-PDZ1 or MBP-PDZ2. *Figure 7G* summarizes these Dlg1-Fz4 binding experiments.

Finally, we have asked whether co-transfection with Dlg1 alters the overall abundance or surface localization of Fz4 (*Figure 7—figure supplement 1*). By immunoblotting of Rim-Fz4 following surface biotinylation, we observed no effect of Dlg1 over-expression on total or surface accumulation of Fz4. In a parallel experiment with Tspan12, the total abundance of Tspan12 was unaltered by Dlg1 over-expression and cell surface Tspan12 was below the limit of detection either with or without Dlg1. These data indicate that Dlg1, while capable of interacting with the Fz4 C-terminus, does not enhance Fz4 trafficking to the plasma membrane.

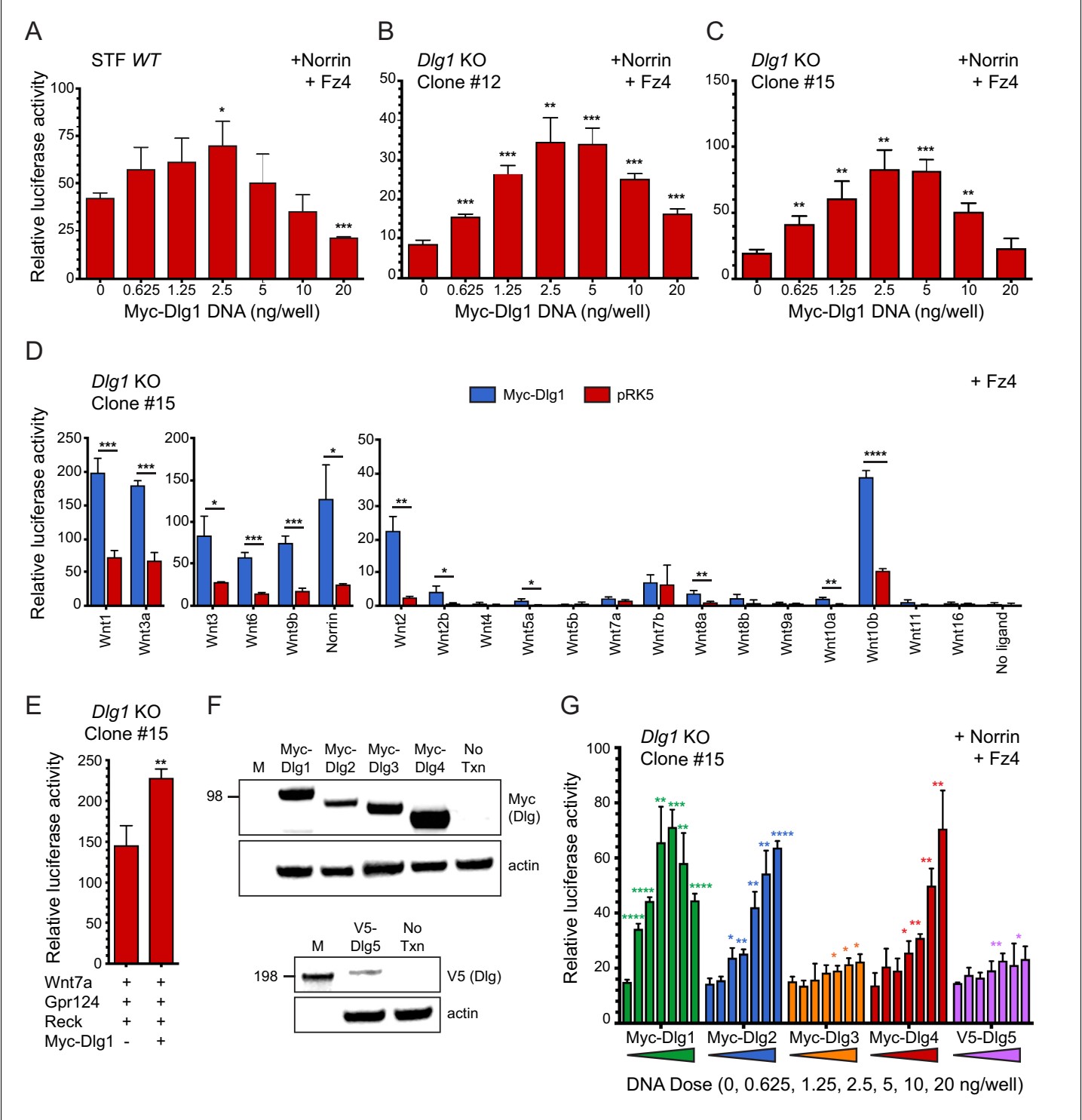

**Figure 6.** Dlg1 activates Wnt signaling in transfected cells. (**A–C**) *WT* STF (**A**), *Dlg1* KO STF Clone #12 (**B**), and *Dlg1* KO STF Clone #15 (**C**) were transfected with Norrin and Fz4 plasmids together with the indicated concentrations of Myc-Dlg1 plasmid DNA. The statistical significance of the relative STF firefly luciferase activity (i.e. normalized relative to the transfected Renilla luciferase) was in comparison to the sample with no Myc-Dlg1 transfection, determined from triplicate data using the unpaired t-test. The mean Renilla luciferase values (n = 42 data points per cell line) were 4.54 ± 0.57 and 3.61 ± 0.92 for Clone #12 and Clone #15, respectively, indicating on average an ~25% greater transfection efficiency for Clone #12 compared to Clone #15. (**D**) *Dlg1* KO STF Clone #15 was transfected with Fz4 and each of the 19 mouse Wnts or Norrin, together with Myc-Dlg1 or the empty pRK5 vector at 2.5 ng/well. (**E**) *Dlg1* KO STF Clone #15 was transfected with Fz4, Wnt7a, Gpr124, and Reck, together with Myc-Dlg1 or the empty pRK5 vector at 2.5 ng/well. (**F**) Immunoblot analysis of Myc-Dlg1, Myc-Dlg2, Myc-Dlg3, Myc-Dlg4, or V5-Dlg5 produced by transient transfection of

*Figure 6 continued on next page*

*Figure 6 continued*

HEK/293T cells. Actin serves as a loading control. Molecular weight markers are indicated at the left. (**G**) *Dlg1* KO STF Clone #15 was transfected with Norrin and Fz4 plasmids with the indicated concentrations of Myc-Dlg1, Myc-Dlg2, Myc-Dlg3, Myc-Dlg4, or V5-Dlg5 plasmid DNA.

DOI: https://doi.org/10.7554/eLife.45542.010

The following figure supplement is available for figure 6:

**Figure supplement 1.** *Dlg1* null mutations created by CRISPR/Cas9-mediated gene editing in the STF reporter cell line.

DOI: https://doi.org/10.7554/eLife.45542.011

In summary, using multiple approaches, we confirm specific binding between the PDZ domains of Dlg1 and the C-terminus of Fz4. For this interaction, the relative importance of the Dlg1 PDZ domains is PDZ2 >PDZ1>>PDZ3.

## Dlg1 stimulation of beta-catenin signaling and retinal vascular development does not require binding to the Fz4 C-terminus

To determine the domains in Dlg1 that are important for stimulating beta-catenin signaling, we generated a series of domain deletion mutants in Myc-Dlg1 in addition to the ones described above in the PDZ domains (*Figure 8—figure supplement 1*). By immunoblotting, all of the deletion mutants were produced at levels comparable to WT Myc-Dlg1 (*Figure 7D and E*; *Figure 8—figure supplement 1C*), and many retained substantial activity in stimulating *Dlg1* KO STF cells transfected with Norrin, Fz4, and Tspan12 (*Figure 8—figure supplement 1B*). For example, deletion of L27+PEST (mutant #1), PDZ1 (mutant #2), PDZ2 (mutant #3), or PDZ1+2 (mutant #5) domains reduced Dlg1 activity by ~25%. All of the mutants that include a deletion of PDZ3 (mutant #4, #6, and #7) reduced activity by ~30–50%. The greatest loss of function was observed upon deletion of the C-terminal SH3+Hook+guanylate kinase (GUK) domains (mutant #11), with modest decreases upon deletion of either SH3+Hook (mutant #8) or Hook+GUK (mutant #10). These data imply that multiple domains within Dlg1 contribute to its activity.

The modest effect of PDZ1+2 deletion on Dlg1 activity in STF cells, led us to ask whether mutating the C-terminal PDZ-binding motif of Fz4 from ETVV to AAAA [Fz4(4A)] affected stimulation by co-transfected Dlg1. A comparison of the luciferase response of *Dlg1* KO STF cells to transfection with Norrin and WT Fz4 vs. Fz4(4A), together with variable amounts of Myc-Dlg1 plasmid, showed that Fz4(4A) elicited a ~5 fold lower response than WT Fz4 (*Figure 8—figure supplement 1D*), roughly consistent with the several-fold lower level of Fz4(4A) accumulation, as determined by immunoblotting (*Figure 8C*). Importantly, the combination of Norrin and Fz4(4A) exhibited a similar Myc-Dlg1 dose dependence – especially at low Myc-Dlg1 plasmid concentrations – as did the combination of Norrin and WT Fz4, suggesting that Dlg1 stimulates beta-catenin signaling in the same manner in these two contexts. In conclusion, the Dlg1-dependent stimulation of beta-catenin signaling activity by a Fz4 mutant that abrogates Dlg1-Fz4 binding argues that this interaction is dispensable for beta-catenin signaling, at least in STF reporter cells.

Signaling assays in cultured cells that are based on over-produced proteins represent only a rough approximation to the situation in vivo, where low levels of signaling proteins, the presence of additional regulatory components, and cell-type specific patterns of gene expression can dramatically alter cellular responses. Therefore, we sought to test the potential relevance of the Dlg1-Fz4 interaction in vivo by using CRISPR/Cas9 editing to create a *Fz4* allele with a mutated C-terminus that is unable to bind to Dlg1. For this experiment, the C-terminal three codons of *Fz4* were mutated from TVV to MGK, hereafter referred to as *Fz4*$^{MGK}$ (*Figure 8A*). [The *Fz4*$^{MGK}$ allele was obtained as a fortuitous in-frame non-homologous end joining (NHEJ) event in an experiment in which we failed to obtain a *Fz4*$^{4A}$ allele by homology-directed repair (HDR).] As would be predicted based on structure-function analyses of the interaction between PDZ-binding motifs and the Dlg1 PDZ domains (*Zhang et al., 2011*; *Slep, 2012*), the TVV-to-MGK substitution eliminates binding to Myc-Dlg1, as determined by the biotin-peptide binding assay (*Figure 8B*). We also observed that Fz4(MGK), like Fz4(4A), accumulates in transfected HEK/293T cells to a level that is several-fold lower than WT Fz4 (*Figure 8C*).

At P18, *Fz4*$^{MGK/MGK}$ mice have normal retinal vascular architecture and density, retinal ECs are uniformly Claudin5+/PLVAP-, and the BRB is intact as judged by the absence of sulfo-NHS-biotin

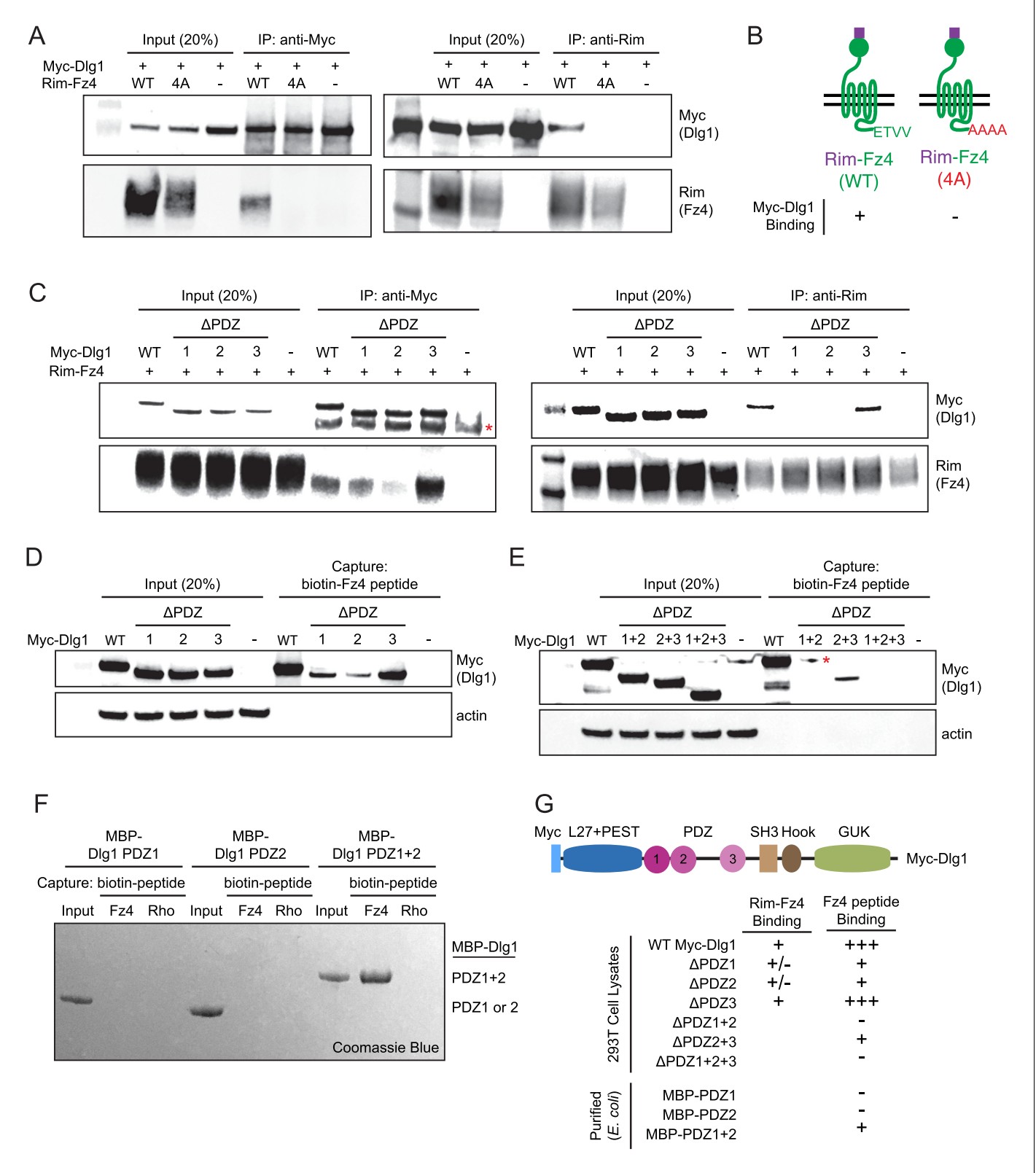

**Figure 7.** The Fz4 C-terminal PDZ-binding motif binds to PDZ domains 1 and 2 of Dlg1. (**A**) Co-immunoprecipitation followed by immunoblotting to assess the interactions between Myc-Dlg1 and Rim-Fz4. HEK/293T cells were co-transfected with Myc-Dlg1 and WT Rim-Fz4, Rim-Fz4(4A), or an empty expression vector. Co-immunoprecipitation was performed with anti-Myc antibody (left) or anti-Rim antibody (right). (**B**) Schematic summarizing the results in (**A**). Rim-Fz4 is depicted in the plasma membrane (horizontal black lines) with its Rim-tagged N-terminus in the extracellular space (top) and its

*Figure 7 continued on next page*

Figure 7 continued

C-terminus in the cytoplasm (bottom). (**C**) HEK/293T cells were co-transfected with Rim-Fz4 and either WT Myc-Dlg1, the indicated PDZ deletion derivatives (ΔPDZ), or an empty expression vector. Co-immunoprecipitation was performed with anti-Myc antibody (left) or anti-Rim antibody (right). In (**C**) and (**E**), the red dot indicates a contaminant, which is variably present. (**D,E**) HEK/293T cells were transfected with WT Myc-Dlg1, the indicated PDZ deletion derivatives (ΔPDZ), or an empty expression vector. Soluble cytosolic proteins from the transfected cells were captured with an N-terminally biotinylated synthetic peptide corresponding to the C-terminal 12 amino acids of Fz4. (**F**) MBP was fused at its C-terminus to PDZ1, PDZ2, or PDZ1+2 of Dlg1 and the resulting fusion proteins were expressed in *E. coli*. The MBP fusion proteins were purified and then tested for binding to an N-terminally biotinylated synthetic peptide corresponding to the C-terminal 12 amino acids of Fz4 or the C-terminal 11 amino acids of bovine rhodopsin, which lacks a PDZ-binding motif. (**G**) Summary of the binding results between Dlg1 and Fz4. For each interaction examined, a score of - (no binding), +/- (little to no binding), + (minimal binding), or +++ (robust binding) was assigned. Blank spaces signify that binding was not assessed.
DOI: https://doi.org/10.7554/eLife.45542.012

The following figure supplement is available for figure 7:

**Figure supplement 1.** Plasma membrane localization of Fz4 in *Dlg1* KO STF Cells.
DOI: https://doi.org/10.7554/eLife.45542.013

leakage (*Figure 8D*; quantified in 8G-I). *Fz4^{MGK/MGK}* mice also have normal brain vascular architecture and barrier integrity. Since homozygosity for a *Fz4* null allele produces a severe retinal vascular phenotype and loss of BBB integrity in the cerebellum and olfactory bulb, these data imply that the Fz4^{MGK} protein possesses substantial function in vivo.

If Dlg1 stimulation of beta-catenin signaling is mediated by Dlg1-Fz4 binding, then (i) EC loss of *Dlg1* (*Dlg1^{fl/Δ};Tie2-Cre*) should produce a retinal vascular phenotype that is no more severe than that produced by *Fz4^{MGK/MGK}* and (ii) EC loss of *Dlg1* on a *Fz4^{MGK/MGK}* genetic background (*Fz4^{MGK/MGK};Dlg1^{fl/Δ};Tie2-Cre*) should produce a phenotype that is no more severe than that produced by homozygous loss of *Dlg1* alone. Contrary to the first of these predictions, at P18 *Dlg1^{fl/Δ};Tie2-Cre* retinas show mildly elevated PLVAP expression in ECs and mild sulfo-NHS-biotin leakage (*Figure 8E*; quantified in 8G-I; see also *Figure 1J–L*), neither of which are present in *Fz4^{MGK/MGK}* retinas. Contrary to the second of these predictions, *Fz4^{MGK/MGK};Dlg1^{fl/Δ};Tie2-Cre* retinas show reduced vascular density, decreased Claudin5 and increased PLVAP in ECs, and widespread sulfo-NHS-biotin leakage (*Figure 8F–I*) – phenotypes that are more severe than those of *Fz4^{MGK/MGK}* or *Dlg1^{fl/Δ};Tie2-Cre* retinas. These findings imply that Dlg1 activates beta-catenin signaling and controls retinal angiogenesis and BRB development and maintenance by a mechanism that is independent of its interaction with the C-terminus of Fz4.

## Discussion

The experiments described above provide strong evidence that the scaffolding protein Dlg1 enhances beta-catenin signaling in ECs in the context of CNS angiogenesis and BBB/BRB development and maintenance (*Figure 8—figure supplement 2*). This conclusion is based on: (i) the similarity in the retinal vascular phenotypes associated with EC-specific loss of *Dlg1* and EC-specific reductions in beta-catenin signaling, (ii) the phenotypic synergy produced by the combination of EC-specific loss of *Dlg1* and loss-of-function mutations in Fz4 or Tspan12 (in the retina) and Ndp or Tspan12 (in the brain), (iii) the nearly complete rescue of EC-specific loss of *Dlg1* by EC-specific stabilization of beta-catenin, and (iv) the 3–4 fold enhancement of beta-catenin signaling stimulated by multiple Wnts or Norrin in *Dlg1* knockout cell lines in response to Dlg1 transfection. Taken together, these data reveal a new role for a member of the Dlg protein family and they suggest that the phenotypes associated with loss of *Dlg1* in mammals (and *Dlg* in Drosophila) may be, at least in part, referable to defects in beta-catenin signaling. It would be interesting to explore the possibility that additional molecular changes might accompany the loss of *Dlg1* in ECs; such changes might be revealed by comparing the EC transcriptomes of WT and *Dlg1* conditional knockout mice.

A surprising finding is that the binding between the Fz4 C-terminal PDZ-binding motif and the PDZ domains of Dlg1 – which is robustly observed with synthetic peptides and recombinant proteins produced in bacterial or mammalian systems – appears to play little or no role in the Dlg1-dependent enhancement of beta-catenin signaling in CNS ECs. This possibility was initially suggested by the observation that mutation of the Fz4 C-terminal PDZ-binding motif in vivo (Fz4^{MGK}) has no apparent retinal vascular phenotype – that is, it does not recapitulate the phenotype associated with

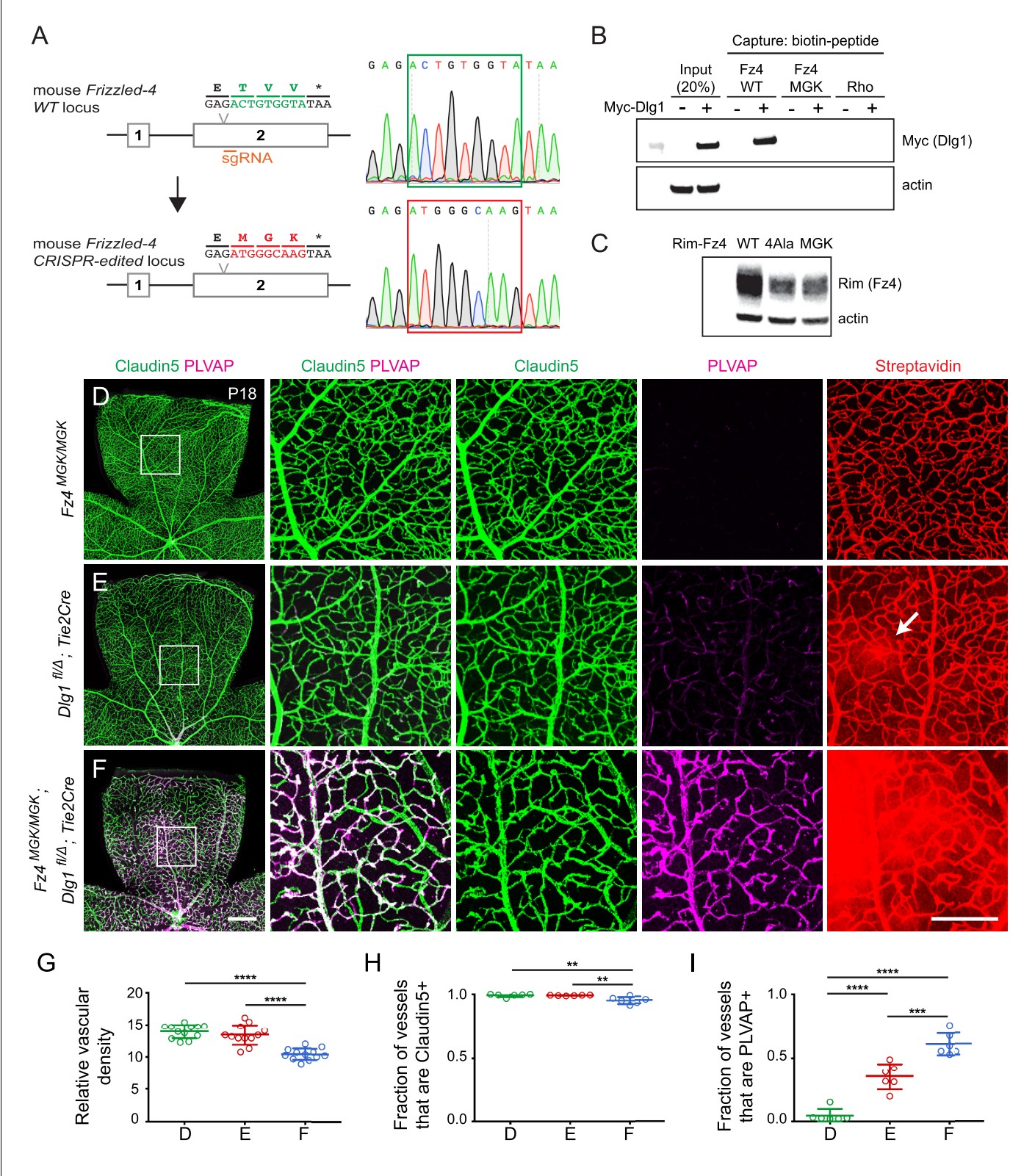

**Figure 8.** Mutation of the Fz4 C-terminal PDZ-binding domain enhances the severity of retinal vascular defects in a *Dlg1* mutant background. (**A**) Diagram of the *Fz4* allele produced by CRISPR/Cas9 editing (*Fz4^MGK*). Numbers indicate exons. TVV, the C-terminal three amino acids, were substituted with MGK. Right, sequencing chromatograms from mouse tail PCR products. Boxes encompass the three altered codons. Top, WT allele. Bottom, *Fz4^MGK* allele. (**B**) Protein extracts from HEK/293T cells transfected with WT Myc-Dlg1 (+) or an empty expression vector (–) were subjected to affinity

*Figure 8 continued on next page*

*Figure 8 continued*

capture using a biotin-Fz4 WT C-terminal peptide, a biotin-Fz4 peptide with the C-terminal three amino acids substituted by MGK, or a biotin-rhodopsin C-terminal peptide. Myc-Dlg1 binds only to the Fz4 WT C-terminal peptide. (C) Immunoblot of protein extracts from HEK/293T cells transfected with Rim-Fz4 WT, Rim-Fz4(4A) or Rim-Fz4(MGK), i.e., Fz4 with the C-terminal three amino acids changed to MGK. (D–F) Maximum projection of the superficial, intermediate, and deep vascular plexuses of P18 retinas from the indicated genotypes (column 1) with boxed regions enlarged in columns 2–5. Mice were injected IP with 2–3 mg of sulfo-NHS-biotin 1–2 hr before sacrifice. Arrow in (E) indicates biotin leakage. Scale bar for column 1, 400 μm. Scale bar for columns 2–5, 200 μm. (G–I) Quantification of summed vascular density (G), the fraction of vessels that immunostain for Claudin5 (H), and the fraction of vessels that immunostain for PLVAP (I), for the genotypes shown in (D–F).

DOI: https://doi.org/10.7554/eLife.45542.014

The following figure supplements are available for figure 8:

**Figure supplement 1.** Effect of Dlg1 domain deletions on canonical beta-catenin signaling in STF cells, and effect of WT Dlg1 on signaling by WT Rim-Fz4 vs. Rim-Fz4(4A).

DOI: https://doi.org/10.7554/eLife.45542.015

**Figure supplement 2.** Summary diagram of Dlg1 in beta-catenin signaling for CNS angiogenesis and BBB/BRB maintenance.

DOI: https://doi.org/10.7554/eLife.45542.016

EC-specific loss of *Dlg1*. It is further supported by the observation that combining Fz4$^{MGK}$ and EC-specific loss of *Dlg1* produces a retinal vascular phenotype that is substantially more severe than that produced solely by EC-specific loss of *Dlg1*. While these observations do not preclude the possibility of an in vivo interaction between Dlg1 and the Fz4 C-terminus, they imply that an interaction between the Fz4 C-terminus and a different partner is of greater functional significance. The most parsimonious explanations for the genetic data are that (i) loss of the Fz4 C-terminal motif reduces beta-catenin signaling in vivo (as it does in transfected cells) but not so much as to produce a retinal vascular phenotype, (ii) loss of Dlg1, which interacts with proteins other than Fz4, reduces beta-catenin signaling in vivo to produce a mild retinal vascular phenotype, and (iii) simultaneous loss of both the Fz4 C-terminal motif and Dlg1 reduces beta-catenin signaling further than with either mutation alone, producing a more severe vascular phenotype.

## Roles of the Dlg family in epithelia and neurons

The first Dlg family member was identified in Drosophila where its loss-of-function phenotype causes neoplastic transformation of imaginal disc cells (*Woods and Bryant, 1991*). In vertebrate epithelial cells, Dlg1 localizes to the basolateral membrane, but its role in epithelial polarity is less clear, as *Dlg1* knockout mice show no obvious epithelial polarity defects (*Iizuka-Kogo et al., 2007*; *Mahoney et al., 2006*), a result that could reflect redundancy with one or more of the four other mammalian Dlg family members (*Yamanaka and Ohno, 2008*).

The best understood role for mammalian Dlg family members is in neurons, where Dlg1 (SAP-97), Dlg2 (PSD-93/Chapsyn-110), Dlg3 (SAP-102), and Dlg4 (PSD-95/SAP-90) control receptor and channel localization and trafficking, and loss or over-expression of Dlg family members alters synaptic transmission and/or synaptic plasticity (*Xu, 2011*; *Fourie et al., 2014*; *Won et al., 2017*). In the adult mouse brain, Dlg proteins are expressed in astrocytes, neurons, oligodendrocytes, microglia, and ECs (*Zhang et al., 2014*). In addition to their roles in neurons, accumulating evidence implicates Dlg1 and other Dlg family members in protein trafficking and localization in non-neuronal contexts (*Walch, 2013*).

While our data support a role for Dlg1 in stimulating beta-catenin signaling in CNS ECs, further studies will be needed to explore this function in other cell types. The STF data presented here imply that Dlg2 and Dlg4 also have beta-catenin stimulatory activity (*Figure 6G*). Whether these Dlg family members similarly activate beta-catenin signaling in vivo has yet to be determined.

## Expanding the biological repertoire of the Dlg family

In the context of CNS ECs, the data presented here imply that Dlg1 enhances beta-catenin signaling by interacting with one or more proteins other than Fz4. In the paragraphs that follow, we discuss candidate interactions that might mediate this activity, with a focus on PDZ interactions. We note, however, that multiple domains of Dlg1 are likely to be relevant, based on the domain deletion experiments in *Figure 8—figure supplement 1A–C*.

One obvious candidate for interacting with Dlg1 in the context of beta-catenin signaling is the Adenomatous Polyposis Coli (APC) protein, which has a C-terminal PDZ-binding motif that binds to the PDZ1 and PDZ2 domains of Dlg1 with affinities of 18 uM and 1 uM, respectively (*Zhang et al., 2011*; *Sotelo et al., 2012*). The corresponding atomic resolution structures of these two peptide-PDZ complexes show the canonical mode of binding (*Zhang et al., 2011*; *Slep, 2012*). APC is a large (~300 kDa) protein that plays an important role in assembling the beta-catenin destruction complex (*Munemitsu et al., 1995*; *Rubinfeld et al., 1996*; *Stamos and Weis, 2013*). Other APC functions include modulating cytoskeletal dynamics and cytoskeletal signaling. In addition to binding to Dlg1, APC binding partners include the kinesin KAP3 (*Jimbo et al., 2002*), the GTPase activating protein IQGAP (*Watanabe et al., 2004*), the guanine nucleotide exchange factor ASEF (*Kawasaki et al., 2000*), and the plus ends of polymerizing microtubules (*Bahmanyar et al., 2009*). We speculate that Dlg1-mediated changes in the subcellular localization of APC or in the interaction of APC with its protein partners could reduce APC-dependent beta-catenin destruction, thereby enhancing beta-catenin signaling. We note, however, that this interaction depends on the PDZ1 and PDZ2 domains of Dlg1, deletion of which reduces but does not eliminate beta-catenin stimulation in STF cells (*Figure 8—figure supplement 1A and B*).

A second candidate for interacting with Dlg1 in the context of beta-catenin signaling in CNS ECs is Gpr124, a seven transmembrane protein in the adhesion GPCR family that functions in concert with the GPI-anchored protein Reck to enhance Wnt7a/b-specific signaling through Frizzled receptors and Lrp5/6 co-receptors (*Zhou and Nathans, 2014*; *Posokhova et al., 2015*; *Vanhollebeke et al., 2015*; *Cho et al., 2017*; *Eubelen et al., 2018*; *Vallon et al., 2018*). Gpr124 plays a major role in angiogenesis and vascular barrier development in the brain (*Kuhnert et al., 2010*; *Anderson et al., 2011*; *Cullen et al., 2011*) and a minor role in angiogenesis and vascular barrier development in the retina (*Wang et al., 2018a*). As noted in the Results section, Gpr124 has a C-terminal PDZ-binding motif that has been shown to bind to Dlg1 (*Yamamoto et al., 2004*; *Posokhova et al., 2015*). However, deletion of the Gpr124 PDZ binding motif or most of the Gpr124 intracellular domain impairs beta-catenin signaling by only ~20–40% in transfected cells (*Posokhova et al., 2015*; *Vallon et al., 2018*), and Dlg1 co-transfection produces only a ~1.5x stimulation of Wnt7a/Fz/Gpr124/Reck signaling in STF cells (*Figure 6E*). Whether there is a larger synergy among these components in vivo remains to be determined.

By analogy with the *Dlg1* and *Fz4* genetic interaction experiments described here, it would be of interest to examine genetic interactions between *Dlg1* and other *Frizzled* genes. Late gestation *Dlg1*$^{-/-}$ fetuses exhibit (i) mis-orientation of the stereociliary bundles in inner ear sensory hair cells, (ii) defects in the closure of the palate, neural tube, and eyelids, and (iii) multiple cardiac defects, including ventricular septal defect, persistent truncus arteriosus, and double outlet right ventricle (*Rivera et al., 2013*; *Iizuka-Kogo et al., 2015*). In various combinations, these defects are precisely the ones present in the following double mutant fetuses: *Fz3*$^{-/-}$;*Fz6*$^{-/-}$, *Fz1*$^{-/-}$;*Fz2*$^{-/-}$ and *Fz2*$^{-/-}$;*Fz7*$^{-/-}$ (*Wang et al., 2006*; *Yu et al., 2010*; *Yu et al., 2012*). The similarities between *Dlg1*$^{-/-}$ and these Frizzled double mutant phenotypes suggest that they result from disruptions in the same signaling pathway. More specifically, the *Dlg1*$^{-/-}$ phenotypes have been interpreted as reflecting a defect in planar cell polarity signaling (*Rivera et al., 2013*; *Iizuka-Kogo et al., 2015*). This interpretation could account for those phenotypes referable to reduced signaling by Fz3 and Fz6, two Frizzleds that appear to be devoted exclusively to planar cell polarity signaling, as judged by their asymmetric subcellular localization and their failure to activate beta-catenin (i.e. canonical Wnt) signaling in cell culture (*Wang et al., 2006*; *Yu et al., 2012*). However, this interpretation may be incomplete for those phenotypes referable to reduced signaling by Fz1, Fz2, and Fz7, as current evidence suggests that this Frizzled subfamily can transduce both planar polarity and beta-catenin signals (*Lhomond et al., 2012*; *Yu et al., 2012*). Additionally, *Dlg1*$^{-/-}$ fetuses exhibit defects in renal and urogenetial development that are partially overlapping with defects caused by knockout of *Wnt4*, further suggestive of a link to beta-catenin signaling (*Vainio et al., 1999*; *Iizuka-Kogo et al., 2007*). In light of the evidence presented here that Dlg1 enhances beta-catenin signaling in CNS ECs, it seems reasonable to suggest that Dlg1 enhances beta-catenin signaling in other biological contexts and that part of the *Dlg1*$^{-/-}$ phenotype reflects this activity.

# Materials and methods

### Key resources table

| Reagent type (species) or resource | Designation | Source or reference | Identifiers | Additional information |
|---|---|---|---|---|
| Genetic reagent (*M. musculus*) | *Dlg1*$^{fl}$ | PMID: 18842882 | | Richard Huganir (Johns Hopkins) |
| Genetic reagent (*M. musculus*) | *Dlg1*$^{\Delta}$ | this paper | | Generated by breeding mice carrying the germline *Sox2-Cre* transgene with floxed mice |
| Genetic reagent (*M. musculus*) | *Fz4*$^{\Delta}$ | PMID: 15035989 | | |
| Genetic reagent (*M. musculus*) | *Fz4*$^{MGK}$ | this paper | | Please find details under Materials and methods (Gene Targeting) |
| Genetic reagent (*M. musculus*) | *Tspan12*$^{\Delta}$ | PMID: 19837033 | | Harald Junge (University of Colorado Boulder) |
| Genetic reagent (*M. musculus*) | *Ndp*$^{\Delta}$ | PMID: 19837032; Jackson Laboratory | Stock #: 012287; RRID:IMSR_JAX:012287 | |
| Genetic reagent (*M. musculus*) | *Ctnnb1*$^{flex3}$ | PMID: 10545105 | | Makoto Taketo (Kyoto University) |
| Genetic reagent (*M. musculus*) | *Tie2-Cre* | Jackson Laboratory | Stock #: 008863; RRID:IMSR_JAX:008863 | |
| Genetic reagent (*M. musculus*) | *Pdgfb-CreER* | PMID: 18257043 | | |
| Genetic reagent (*M. musculus*) | *Sox2-Cre* | PMID: 12617844 | | |
| Cell line (*H. sapiens*) | HEK/293T | ATCC | Cat. #: CRL-3216 | |
| Cell line (*E. coli*) | BL21 | New England Biolabs | Cat. #: C2530H | |
| Cell line (*H. sapiens*) | Super TOP Flash (STF) luciferase reporter cell line | PMID: 15035989 | | |
| Cell line (*H. sapiens*) | STF *Dlg1* KO Clone #12 | this paper | | Please find details under Materials and methods (Constructing CRISPR/Cas9 STF cell lines) |
| Cell line (*H. sapiens*) | STF *Dlg1* KO Clone #15 | this paper | | Please find details under Materials and methods (Constructing CRISPR/Cas9 STF cell lines) |
| Antibody | Rabbit polyclonal anti-Glut1 | Thermo Fisher Scientific | Cat. #: RB-9052-P1; RRID: AB_177895 | 1:400 dilution |
| Antibody | Rat monoclonal anti-PLVAP/MECA-32 | BD Biosciences | Cat. #: 553849; RRID: AB_395086 | 1:400 dilution |
| Antibody | Mouse monoclonal anti-Claudin5, Alexa 488 conjugate | Thermo Fisher Scientific | Cat. #: 352588; RRID: AB_2532189 | 1:400 dilution |
| Antibody | Rabbit polyclonal anti-6xMyc | PMID: 28803732 | | 1:10,000 dilution |
| Antibody | Rabbit monoclonal anti-V5 | Cell Signaling Technology | Cat. #: 13202; RRID: AB_2687461 | 1:1000 dilution |
| Antibody | Mouse monoclonal anti-RIM3F4 | PMID: 9092582 | | 1:10,000 dilution |

*Continued on next page*

*Continued*

| Reagent type (species) or resource | Designation | Source or reference | Identifiers | Additional information |
|---|---|---|---|---|
| Antibody | Mouse monoclonal anti-actin | Millipore Sigma | Cat. #: MAB1501; RRID: AB_2223041 | 1:10,000 dilution |
| Antibody | Rabbit monoclonal anti-SAP97 | Abcam | Cat. #: ab134156 | 1:1000 dilution |
| Antibody | Goat polyclonal anti-rabbit IgG (H + L) cross-adsorbedsecondary antibody, Alexa 488, 594, and 647 conjugates | Thermo Fisher Scientific | Cat. #s: A-11008, RRID: AB_143165; A-11012, RRID: AB_2534079; A-21244,RRID: AB_2535812 | 1:400 dilution |
| Antibody | IRDye 800CW goat anti-mouse IgG (H + L) secondary antibody | LI-COR | Cat. #: 925–32210 | 1:10,000 dilution |
| Antibody | IRDye 680RD goat anti-rabbit IgG (H + L) secondary antibody | LI-COR | Cat. #: 925–68071 | 1:10,000 dilution |
| Oligonucleotides | *Fz4^{MGK}* guide RNA: AGGAAAAG GCAACGAGACTG | this paper | | Please find details under Materials and methods (Gene Targeting) |
| Oligonucleotides | *Dlg1* guide RNA: AAGCTCAT TAAAGGTCCTAA | this paper | | Please find details under Materials and methods (Constructing CRISPR/Cas9 STF cell lines) |
| Recombinant DNA reagents | Rat Myc-Dlg1 cDNA | PMID: 12805297 | | Richard Huganir (Johns Hopkins) |
| Recombinant DNA reagents | pMAL-cR1 expression vector | New England Biolabs | | |
| Recombinant DNA reagents | Mouse Norrin, Wnts, and Frizzleds cDNA | PMID: 23095888 | | |
| Recombinant DNA reagents | Mouse Tspan12 cDNA | PMID: 30478038 | | |
| Recombinant DNA reagents | Mouse Reck cDNA | PMID: 28803732 | | |
| Recombinant DNA reagents | Mouse Gpr124 cDNA | PMID: 28803732 | | |
| Recombinant DNA reagents | Mouse Dlg2 cDNA | Origene | Cat. #: MR222602 | |
| Recombinant DNA reagents | Mouse Dlg3 cDNA | GE Dharmacon | Clone #: 6842105 | |
| Recombinant DNA reagents | Mouse Dlg4 cDNA | GE Dharmacon | Clone #: 4501403 | |
| Recombinant DNA reagents | Mouse V5-Dlg5 cDNA | PMID: 17765678 | | Alex Kolodkin (Johns Hopkins) |
| Recombinant DNA reagents | pSpCasp9(BB) −2A-GFP vector | Addgene | Plasmid #: 48138 | |
| Peptide, recombinant protein | Rhodopsin peptide (SKTETSQVAPA) | this paper | | Please find details under Materials and methods (Biotin-peptide binding to proteins expressed in HEK/293T cells) |
| Peptide, recombinant protein | Fz4 WT peptide (WVKPGKGNETVV) | this paper | | Please find details under Materials and methods (Biotin-peptide binding to proteins expressed in HEK/293T cells) |

*Continued on next page*

*Continued*

| Reagent type (species) or resource | Designation | Source or reference | Identifiers | Additional information |
|---|---|---|---|---|
| Peptide, recombinant protein | Fz4 MGK peptide (WVKPGKGNEMGK) | this paper | | Please find details under Materials and methods (Biotin-peptide binding to proteins expressed in HEK/293T cells) |
| Commercial assay or kit | MEGAshortscript T7 Kit | Invitrogen | Cat. #: AM1354 | |
| Commercial assay or kit | MEGAclear Transcription Clean-Up Kit | Invitrogen | Cat. #: AM1908 | |
| Commercial assay or kit | mMESSAGE mMACHINE T7 Ultra Transcription Kit | Invitrogen | Cat. #: AM1345 | |
| Commercial assay or kit | Dual-Luciferase Reporter Assay System | Promega | Cat. #: E1910 | |
| Chemical compound, drug | (Z)—4-Hydroxytamoxifen | Sigma-Aldrich | Cat. #: H7904 | |
| Chemical compound, drug | Sunflower seed oil from *Helianthus annuus* | Sigma-Aldrich | Cat. #: S5007 | |
| Chemical compound, drug | EZ-Link Sulfo-NHS-LC-Biotin | Thermo Fisher Scientific | Cat. #: 21335 | |
| Software, algorithm | ImageJ | https://imagej.nih.gov/ij | | |
| Software, algorithm | Adobe Photoshop CS6 | https://adobe.com/photoshop | | |
| Software, algorithm | Adobe Illustrator CS6 | https://adobe.com/illustrator | | |
| Software, algorithm | GraphPad Prism 7 | http://www.graphpad.com | | |
| Other | FuGENE HD Transfection Reagent | Promega | Cat. #: E2311 | |
| Other | Pierce NeutrAvidin agarose resin | Thermo Fisher Scientific | Cat. #: 29200 | |
| Other | Protein G Dynabeads | Thermo Fisher Scientific | Cat. #: 10004D | |
| Other | Amylose resin | New England Biolabs | Cat. #: E8021S | |
| Other | Streptavidin Dynabeads | Thermo Fisher Scientific | Cat. #: 11047 | |
| Other | Fluoromount G | EM Sciences | Cat. #: 17984–25 | |
| Other | Protease Inhibitor | Roche | Cat. #: 11836170001 | |
| Other | 1x BugBuster Protein Extraction Reagent | Millipore Sigma | Cat. #: 70584 | |
| Other | Texas Red Streptavidin | Vector Laboratories | Cat. #: SA-5006 | |

## Gene targeting

The *Fz4$^{MGK}$* mouse was generated using CRISPR/Cas9 gene editing. An sgRNA (AGGAAAAGG-CAACGAGACTG) targeting exon 2 of *Fz4* was selected and synthesized, as described by *Pelletier et al. (2015)*. Briefly, a dsDNA template containing the sequences for a T7 promoter, the sgRNA, and a tracrRNA scaffold was generated by tandem PCR amplification and purification on a Qiagen column (Qiagen 28106). In vitro transcription of this dsDNA template was performed using the MEGAshortscript T7 Kit (Invitrogen AM1354), and products were purified using the MEGAclear Transcription Clean-Up Kit (Invitrogen AM1908). The Cas9 mRNA was transcribed from a modified

pX330 plasmid (Addgene 42230), which contains a T7 promoter (pX330+T7), using the mMESSAGE mMACHINE T7 Ultra Transcription Kit (Invitrogen AM1345). The transcript was purified by LiCl precipitation. The quality of all purified transcripts was confirmed using an Agilent 2100 Bioanalyzer. The sgRNA and Cas9 mRNA were injected into C57BL/6 x SJL F2 embryos and correctly targeted founders were identified by PCR and by sequencing of cloned PCR products.

## Mice

The following mouse alleles were used: $Dlg1^{fl}$ (*Zhou et al., 2008*), $Fz4^{\Delta}$ (*Xu et al., 2004*); $Tspan12^{\Delta}$ (*Junge et al., 2009*); $Ndp^{\Delta}$ (JAX 012287; *Ye et al., 2009*); $Ctnnb1^{flex3}$ (*Harada et al., 1999*), *Tie2-Cre* (JAX 008863), and *Pdgfb-CreER* (*Claxton et al., 2008*). The $Dlg1^{\Delta}$ allele was generated by breeding mice carrying a germline *Sox2-Cre* transgene (*Hayashi et al., 2002*) with $Dlg1^{fl}$ mice. All mice were housed and handled according to the approved Institutional Animal Care and Use Committee (IACUC) protocol MO16M369 of the Johns Hopkins Medical Institutions.

## Antibodies and other reagents

The following antibodies were used for tissue immunohistochemistry: rabbit anti-GLUT1 (Thermo Fisher Scientific RB-9052-P1); rat anti-mouse PLVAP (MECA-32; BD Biosciences 553849); mouse anti-CLDN5, Alexa Fluor 488 conjugate (Thermo Fisher Scientific 352588); Texas Red streptavidin (Vector Laboratories SA-5006). Alexa Fluor-labeled secondary antibodies were from Thermo Fisher Scientific.

The following antibodies were used for immunoblot analysis: rabbit anti-6xMyc (JH6204); rabbit mAb anti-V5 (Cell Signaling 13202S); mouse mAb anti-RIM3F4 (ascites); mouse anti-actin (Millipore Sigma MAB1501); rabbit anti-SAP97 (Abcam ab134156). Fluorescent secondary antibodies for immunoblotting were from Li-Cor.

## Tissue processing and immunohistochemistry

For retinal vasculature analysis, whole-mount retinas were prepared and processed for immunohistochemical analysis as described by *Wang et al. (2018a)*. Briefly, mouse eyes were harvested and immersion fixed for one hour at room temperature (RT) with 1% PFA in PBS. Whole-mount retinas were dissected and incubated overnight with primary antibodies (1:400) or Texas Red streptavidin (1:100) diluted in 1x PBSTC (1x PBS + 0.5% Triton X-100 + 0.1 mM $CaCl_2$) + 10% normal goat serum (NGS). Retinas were washed at least 3 times with 1x PBSTC over the course of 6 hr, and subsequently incubated overnight with secondary antibodies (1:400) diluted in 1x PBSTC + 10% NGS. The next day, retinas were washed at least 3 times with 1x PBSTC over the course of 6 hr, and flat-mounted using Fluoromount G (EM Sciences 17984–25).

For brain vasculature analysis, brains were prepared and processed for immunohistochemical analysis as described by *Wang et al. (2018a)*. Deeply anesthetized mice were perfused via the cardiac route with 1% PFA in PBS. Dissected brains were dehydrated in 100% MeOH overnight at 4°C. Brains were re-hydrated the following day in 1x PBS at 4°C for at least 3 hr before embedding in 3% agarose. Tissue sections of 150–200 µm thickness were cut using a vibratome (Leica). Sections were incubated overnight with primary antibodies (1:400) or Texas Red streptavidin (1:100) diluted in 1x PBSTC + 10% NGS. Sections were washed at least 3 times with 1x PBSTC over the course of 6 hr, and subsequently incubated overnight with secondary antibodies (1:400) diluted in 1x PBSTC +10% NGS. The next day, sections were washed at least 3 times with 1x PBSTC over the course of 6 hr, and flat-mounted using Fluoromount G (EM Sciences 17984–25).

All antibody incubation and washing steps were performed at 4°C unless otherwise noted. Whole-mount retinas and brain sections were imaged using a Zeiss LSM700 confocal microscope, and processed with ImageJ, Adobe Photoshop, and Adobe Illustrator software.

## 4-HT preparation and administration

4-HT was prepared as described by *Zhou and Nathans (2014)*. In brief, solid 4-HT (Sigma-Aldrich H7904) was dissolved in an ethanol:sunflower seed oil (Sigma-Aldrich S5007) mixture (1:5 vol) to a final concentration of 2 mg/ml and stored in aliquots at −80°C. All injections were performed intraperitoneally. Pups were injected with 25 µl of 4-HT at P0 and again at P1.

## Sulfo-NHS-biotin preparation and administration

Sulfo-NHS-Biotin (Thermo Scientific 21335) was dissolved in 1x PBS to a final concentration of 20 mg/ml. Pups were injected with 150 µl sulfo-NHS-biotin IP 1–2 hr before sacrifice.

## Plasmids

The Myc-Dlg1 expression construct (*Rumbaugh et al., 2003*) was used as a backbone to generate ΔL27+PEST (Δ amino acids (Δaa)4–185), ΔPDZ1 (Δaa187-279), ΔPDZ2 (Δaa282-372), ΔPDZ3 (Δaa429-514), ΔPDZ(1+2) (Δaa187-373), ΔPDZ(2+3) (Δaa282-514), ΔPDZ(1+2+3) (Δaa187-514), ΔSH3+Hook (Δaa548-669), ΔHook (Δaa634-669), ΔHook+GUK (Δaa634-878), and ΔSH3+Hook+GUK (Δaa548-878). A 13 amino acid spacer (GGPGSGGSGAPGG) replaced each of these deleted segments.

The following inserts for MBP fusion constructs were PCR amplified from the Myc-Dlg1 expression vector: PDZ1 (aa187-284), PDZ2 (aa280-374), and PDZ(1+2) (aa187-374). Inserts were sub-cloned into the pMAL-cR1 expression vector.

The mouse Fz4 and Tspan12 expression constructs were used as backbones to generate the Rim-Fz4 and Rim-Tspan12 expression constructs, respectively. Specifically, a Rim epitope tag (NETYDLPLHPRTAG) was inserted after the Fz4 signal sequence and before the Fz4 CRD, and after the Tspan12 signal sequence. Mutant Rim-Fz4 constructs, in which the C-terminal four amino acids (ETVV) were converted to 4A (AAAA) or MGK, were generated by PCR.

The mouse Dlg2 (Origene MR222602), mouse Dlg3 (GE Dharmacon; clone 6842105), and mouse Dlg4 (GE Dharmacon; clone 4501403) cDNA sequences were cloned into the pRK5 mammalian expression vector with an N-terminal Myc epitope (EQKLISEEDL) tag. The V5-Dlg5 plasmid was a kind gift from Dr. Alex Kolodkin (Johns Hopkins).

Expression plasmids for mouse Wnts and Frizzleds (*Yu et al., 2012*), mouse Tspan12 (*Wang et al., 2018a*), and mouse Reck and Gpr124 (*Cho et al., 2017*) have been previously described.

## Cell lines

HEK/293T cells (ATCC CRL-3216) and Super TOP Flash (STF) cells (*Xu et al., 2004*) were used in this study, and there was no evidence of mycoplasma contamination. We confirmed cell line identity by RNA sequencing. Cells were grown in DMEM/F-12 supplemented with 10% fetal bovine serum (FBS) and passaged at a dilution of 1:5 for no more than a maximum of 20 passages. HEK/293T cells were seeded into 6-well or 12-well plates at a confluency of 70–80% prior to transfection. STF cells were seeded into 96-well plates at a confluency of 30–40% prior to transfection. Experimental details are further elaborated below in 'Luciferase assays', 'Co-immunoprecipitation assays', and 'Cell surface biotinylation assays'.

## Constructing CRISPR/Cas9 STF cell lines

The *Dlg1* KO STF cell lines were generated using CRISPR/Cas9 technology. An sgRNA (AAGCTCA TTAAAGGTCCTAA) targeting exon 10 of human *Dlg1* (*hDlg1*) was cloned into the pSpCasp9(BB)−2A-GFP vector (PX458) (Addgene Plasmid #48138). STF cells were plated on a 10 cm plate at a confluency of 60–70%. The following day, fresh DMEM/F-12 (Thermo Fisher Scientific 12500) supplemented with 10% fetal bovine serum (FBS) was replaced in the 10 cm plate. Three hours later, cells were transfected with the PX458 expression plasmid containing the *hDlg1* sgRNA (8 µg of DNA) using FuGENE HD Transfection Reagent (Promega E2311). 48 hr post-transfection, cells were harvested in 1x PBS supplemented with 0.5% bovine serum albumin (BSA).

Single, viable GFP positive cells were sorted into a 96-well plate containing fresh DMEM/F-12 + 10% FBS using a MoFlo XDP Sorter (Beckman Coulter). Single cell clones were expanded and screened for editing of the *hDlg1* locus by genomic PCR and restriction enzyme digestion with *Ppu*M I. All clones lacking a PpuM I cleavage site, which straddles the Cas9 cleavage site, were analyzed by immunoblotting to assess Dlg1 levels (1:1000 rabbit anti-SAP97). Two clones had no detectable Dlg1 (*Dlg1* KO Clones #12 and #15). The targeted region from these two clones was PCR amplified and sub-cloned for Sanger sequencing, which revealed a frame-shift mutation in each *hDlg1* allele.

## Luciferase assays

Dual luciferase assays were performed as described by *Xu et al. (2004)*. Briefly, STF cells were plated on 96-well plates at a confluency of 30–40%. The following day, fresh DMEM/F-12 supplemented with 10% FBS was replaced in each of the wells. Three hours later, cells were transfected in triplicate with expression plasmids (240 ng of DNA per three wells) using FuGENE HD Transfection Reagent. The DNA master mix (unless otherwise specified) included: 1.5 ng of the internal control Renilla luciferase plasmid (pRL-TK), and 60 ng each of the pRK5 expression plasmids for Fz4, Norrin, Tspan12, and/or control vector.

The Myc-Dlg1 expression plasmid was co-transfected either in a dose-dependent manner or at 7.5 ng per three wells. Expression plasmids for domain deletion mutants of Dlg1 were co-transfected at 7.5 ng per three wells. The Myc-Dlg2, Myc-Dlg3, Myc-Dlg4, and V5-Dlg5 were co-transfected in a dose-dependent manner. All expression plasmids for the Wnt ligands were co-transfected at 60 ng per three wells.

48 hr post-transfection, cells were harvested in 1x Passive Lysis Buffer (Promega E194A) for 20 min at room temperature. Lysates were used to measure Firefly and Renilla luciferase activity using the Dual-Luciferase Reporter Assay System (Promega E1910) and a Turner BioSystems Luminometer (TD-20/20). Relative luciferase activity was calculated by normalizing Firefly/Renilla values. GraphPad Prism 7 software was used to generate plots and perform statistical analysis. The mean ± standard deviations are shown.

## Co-immunoprecipitation assays

HEK/293T cells were transfected with pRK5 expression plasmids using FuGENE HD Transfection Reagent. 48 hr post-transfection, cells were lysed in 1x RIPA buffer (50 mM Tris-HCl pH 7.4, 150 mM NaCl, 1% Triton X-100, and 0.5% deoxycholate) containing protease inhibitor (Roche 11836170001). Cell lysates were incubated at 4°C for 30 min and subsequently centrifuged at 10,000xg at 4°C for 20 min to remove cellular debris. Epitope-tagged proteins (i.e. Myc-Dlg1 or Rim-Fz4) were captured by incubating cleared lysates with anti-Myc or anti-Rim antibodies and Protein G Dynabeads (Thermo Fisher Scientific 10004D) overnight at 4°C. Bead were washed 5–6 times with RIPA buffer, and captured proteins, along with input controls, were resolved by SDS-PAGE and transferred to PVDF membranes (Millipore Sigma IPFL00010) for immunoblotting.

To detect immunoprecipitated proteins, membranes were incubated with primary antibodies (1:2,500 mouse anti-Rim or 1:10,000 rabbit anti-6xMyc) with IRDye 680RD Detection Reagent (LiCor 926–68100) diluted in Odyssey Blocking Buffer (LiCor 927–40000) overnight at 4°C. Membranes were washed 3 times with 1x PBS-T (1x PBS + 0.1% Tween 20) and developed using the Odyssey Fc Imaging System (LiCor) to detect immunoprecipitated proteins.

To detect co-immunoprecipitated proteins, membranes were incubated with the reciprocal primary antibodies (1:2,500 mouse anti-Rim or 1:10,000 rabbit anti-6xMyc) diluted in Odyssey Blocking Buffer overnight at 4°C. Membranes were washed 3 times with 1x PBS-T and incubated with LiCor secondary antibodies (1:10,000) diluted in Odyssey Blocking Buffer for 1 hr at room temperature. Membranes were washed at least 3 times with 1x PBS-T and developed using the Odyssey Fc Imaging System.

## Biotin-peptide binding to proteins expressed in HEK/293T cells

The following peptides were chemically synthesized with an N-terminal biotin tag: Rhodopsin peptide (SKTETSQVAPA), Fz4 WT peptide (WVKPGKGNETVV), and Fz4 MGK peptide (WVKPGKGNEMGK). For capturing proteins expressed in HEK/293T cells, NeutrAvidin agarose beads (Thermo Fisher Scientific 29200) were coated with 20 μg of peptide per binding reaction, washed, and incubated with cleared HEK/293T overexpression lysates overnight at 4°C. Resin was washed 5 to 6 times with 1x RIPA buffer, and captured proteins, along with input controls, were resolved by SDS-PAGE and transferred to PVDF membranes for immunoblotting and analysis, as described above.

## Purification and binding of MBP fusion proteins

Expression of MBP fusion proteins was induced in BL21 cells with 1 mM IPTG at an optical density at 600 nm of 0.4–0.6 for 2 hr at 37°C. Cells were pelleted and lysed in 1x BugBuster Protein Extraction

Reagent (Millipore Sigma 70584) for 20 min at 23°C. Cell lysates were centrifuged at 16,000xg at 4°C for 20 min to remove cellular debris.

MBP fusion proteins were batch purified from cleared lysates with amylose resin (New England Biolabs E8021S) in 1x amylose binding buffer (20 mM Tris-HCl pH7.4, 200 mM NaCl, 1 mM EDTA) overnight at 4°C. Resin was washed 10 times with 1x amylose binding buffer with 1% Triton X-100 and 0.5% deoxycholate. Purified MBP fusion proteins were batch eluted in 1x amylose elution buffer (20 mM Tris-HCl pH 7.4, 200 mM NaCl, 1 mM EDTA, 10 mM maltose).

Purified MBP fusion proteins were captured with biotin-peptide coated Streptavidin Dynabeads (Thermo Fisher Scientific 11047) overnight at 4°C in 1x PBS. The next day, Streptavidin Dynabeads were washed 10 times in 1x RIPA buffer, and captured proteins, along with input controls, were resolved by SDS-PAGE and visualized by Coomassie Blue staining.

## Cell surface biotinylation assay

Cell surface biotinylation was performed as described by *Pavel et al. (2014)*. In brief, HEK/293T cells were plated on 0.2% gelatin coated wells and transfected with pRK5 expression plasmids with FuGENE HD Transfection Reagent. The media was removed 48 hr post-transfection, and cells were washed three times with 1x PBS. Cells were incubated in 1x PBS containing Sulfo-NHS-Biotin (250 µg/ml) at 4°C for 30 min. Excess biotin was quenched by adding Tris-HCl pH 7.4 to a final concentration of 50 mM at 4°C for 5 min. After removing the Tris buffer, cells were detached, washed 3 times in 1x Tris buffered saline (TBS), and lysed in 1x RIPA buffer containing protease inhibitor (Roche 11836170001). Cell lysates were incubated at 4°C for 30 min and subsequently centrifuged at 10,000xg at 4°C for 20 min to remove cellular debris. Cell-surface proteins were captured by incubating cleared lysates with NeutrAvidin Agarose Resin (Thermo Fisher Scientific 29200) overnight at 4°C. Resin was washed 5 to 6 times with 1x RIPA buffer, and captured proteins, along with input controls, were resolved by SDS-PAGE and transferred to PVDF membranes (EMD Millipore IPFL00010) for immunoblotting and imaging, as described above.

## Quantification and statistical analysis

For quantifying relative vascular density, whole mount retinas were stained with anti-CLDN5 and anti-PLVAP. Confocal images were scanned at 5 µm intervals along the Z-axis, of which two images were Z-stacked to assess vascular density in the superficial plexus or deep plexus only, or eight images were Z-stacked to assess vascular density in superficial + intermediate + deep plexuses. Images were exported to ImageJ and converted to an 8-bit format. The background (determined with a rolling circle with a 50 pixel radius) was subtracted from all channels. Images were thresholded and skeletonized to pixelate the retinal vasculature. Quantification of relative vascular density was performed by measuring the percentage of pixels covered in a 640 × 640 µm area of the retina. For each biological replicate, four distinct regions of a given Z-stacked image were quantified, and at least three biological replicates for each genotype were analyzed.

For quantifying the fraction of the retina covered by vasculature at P7, images of whole mount retinas (as described above) were exported to ImageJ. All channels were combined into a composite image. The distance from the optic nerve to the peripheral edge of the retinal vasculature was measured as a fraction of the distance from the optic nerve to the peripheral edge of the retina. For each biological replicate, four measurements were made, and at least three biological replicates for each genotype were analyzed.

For quantifying the fraction of vessels that immunostained for CLDN5, PLVAP, or GLUT1, vessels in the designated volume of 320 × 320 × 40 µm (retina) or 1500 × 150 × 100 µm (cerebellum) were manually traced using a Wacom tablet and Adobe Illustrator software. Pixel coverage of the traced image was determined as a fraction of the total designated volume (as described above) for each marker. The vascular density for CLDN5+, PLVAP+, and GLUT1+ vessels was then divided by the total vascular density in the designated volume.

GraphPad Prism 7 software was used to generate plots and to perform statistical analysis. The mean ± standard deviations are shown. Statistical significance was determined by the unpaired t-test, and is represented by * ($p < 0.05$), ** ($p < 0.01$), *** ($p < 0.001$), and **** ($p < 0.0001$).

## Acknowledgements

The authors thank Che Zhang and Harald Junge for the gift of Tspan12$^\Delta$ mice, Richard Huganir for the gift of Dlg1$^{fl}$ mice, and Amir Rattner and Mark Sabbagh for helpful discussions and/or comments on the manuscript. This work was supported by the Howard Hughes Medical Institute (JN), the National Eye Institute (R01 EY018637 to JN), and the Arnold and Mabel Beckman Foundation (JN).

## Additional information

### Competing interests

Jeremy Nathans: Reviewing editor, *eLife*. The other authors declare that no competing interests exist.

### Funding

| Funder | Grant reference number | Author |
| --- | --- | --- |
| Howard Hughes Medical Institute | | Jeremy Nathans |
| National Eye Institute | R01 EY018637 | Jeremy Nathans |
| Arnold and Mabel Beckman Foundation | | Jeremy Nathans |

The funders had no role in study design, data collection and interpretation, or the decision to submit the work for publication.

### Author contributions

Chris Cho, Conceptualization, Data curation, Formal analysis, Validation, Investigation, Visualization, Methodology, Writing—original draft, Writing—review and editing; Yanshu Wang, Data curation, Investigation, Methodology, Optimized and conducted experiments related to tissue harvesting, immunofluorescence, and imaging; Philip M Smallwood, Data curation, Investigation, Optimized and conducted experiments related to molecular cloning and protein purification; John Williams, Data curation, Investigation, Optimized and conducted experiments related to molecular cloning; Jeremy Nathans, Conceptualization, Resources, Formal analysis, Supervision, Funding acquisition, Investigation, Methodology, Writing—original draft, Project administration, Writing—review and editing

### Author ORCIDs

Chris Cho https://orcid.org/0000-0002-0929-6536
Jeremy Nathans https://orcid.org/0000-0001-8106-5460

### Ethics

Animal experimentation: All mice were housed and handled according to the approved Institutional Animal Care and Use Committee (IACUC) protocol MO16M369 of the Johns Hopkins Medical Institutions.

### Decision letter and Author response

Decision letter https://doi.org/10.7554/eLife.45542.019
Author response https://doi.org/10.7554/eLife.45542.020

## Additional files

### Supplementary files

• Transparent reporting form
DOI: https://doi.org/10.7554/eLife.45542.017

**Data availability**

All data generated or analyzed during this study are included in the manuscript and supporting files.

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
