## [Decision Letter]

Thank you for submitting your article "Dlg1 activates beta-catenin signaling to regulate retinal angiogenesis and the blood-retina and blood-brain barriers" for consideration by *eLife*. Your article has been reviewed by three peer reviewers, and the evaluation has been overseen by Joseph Gleeson as the Reviewing Editor and Didier Stainier as the Senior Editor. The reviewers have opted to remain anonymous.

The reviewers have discussed the reviews with one another and the Reviewing Editor has drafted this decision to help you prepare a revised submission.

Summary:

Two reviewers found the manuscript to show strong and convincing evidence of Dlg1 participation in the canonical Wnt pathway and its interaction with fz4, Tspan12. The work provides support for a role of Dlg1 in promoting both Norrin/Fz4/Tspan12 signaling and Wnt7/Fz/Gpr124/Reck signaling, and the rescue by beta-catenin stabilization indicates that this is beta catenin-dependent. The authors provide direct evidence for this interaction in vitro and identify the domains involved. However, based on in vivo evidence obtained generating a new FZD4 mutant line, they conclude that the binding between the Fzd4 C-terminus and the PDZ domains of Dlg1 plays no role in the Dlg1-dependent enhancement of beta-catenin signaling in CNS ECs. Reviewer 2 and 3 found that the work is novel and exciting, experimental design is sound, results and conclusions are convincing and commented that the paper fits very well with the Journal's scope and wide readership. Reviewer 3 commented that the evidence from the Fz4(MGK/MGK) cross is not sufficient to conclude that the Dlg1-Fzd4 interaction plays no role in regulating beta-catenin. Reviewer 1 found that the link between Dlg1 and canonical Wnt pathway unconvincing, and while convinced by the Fz4(MGK/MGK) results, Reviewer 1 encouraged new experiments to directly link Dlg1 with the components of the Wnt pathway.

Essential revisions:

1) The cell culture experiments are confounding, seem to yield insignificant or minor fold change levels, and directly contradict the better experiment of testing Fz4 C terminal modification in vivo, which shows no direct connection between Dlg1 and Fz4. Can the authors reconcile these observations?

2) The last experiment is a good negative result, but besides this specific non-interaction we do not learn anything concrete about Dlg1 in retinal angiogenesis or its connection to signaling pathways. Reviewer 3 encourages the authors to try new experiments to directly link Dlg1 with components of the Wnt pathway and so offer something more than speculation.

3) Without a detailed characterisation of the expression profile of the EC from the mutant mice (which is clearly beyond the scope of this study), it is not possible to confidently conclude that the Dlg4-Fzd4 interaction does not play a role in these phenotypes. We recommend revising the interpretation of the findings to include this possibility in the interpretation of the results. Given the extensive section on possible alternatives to Fzd4 in the Discussion, it would be tempting to ask for some evidence for a role for APC or Gpr124 in this study; however, this is likely to be the focus of following papers.

---

## [Author Response]

Essential revisions:1) The cell culture experiments are confounding, seem to yield insignificant or minor fold change levels, and directly contradict the better experiment of testing Fz4 C terminal modification in vivo, which shows no direct connection between Dlg1 and Fz4. Can the authors reconcile these observations?

We agree that the cell culture experiments that measured the activity of various Dlg1 domain deletions were largely inconclusive since no single domain had an overwhelming effect, which is why these data were relegated to a supplementary figure (now Figure 8—figure supplement 1). However, we disagree with the above comment in regard to the more important cell culture analyses of the activity of Dlg1 and the other Dlg family members in combination with Norrin and Wnts (Figure 6). In Figure 6, we observed that Dlg1 transfection of the Dlg1 KO reporter cell line produced a dose-dependent enhancement of beta-catenin signal strength that was maximally ~4-fold over baseline in the presence of Norrin and various Wnts. A 4-fold change may sound modest, but we note that (i) the reporter cell line also expresses other Dlg family members, which could contribute to the non-zero signal in the absence of Dlg1 transfection (hence lowering the fold induction), (ii) the well-established Norrin/Fz4 signaling co-activator Tspan12 gives only a several-fold induction in this and other reporter assays, and (iii) signaling in transiently transfected reporter cells (where receptors and/or ligands are over-expressed) is an imperfect proxy for signaling in the living animal where all of the signaling proteins are present at low abundance. We specifically disagree with the implication that the changes shown in Figure 6 are: (i) “insignificant” (on the contrary, they are of high statistical significance, and the mouse Dlg1 KO phenotype implies that they are also of biological significance), or (ii) “minor” (see the preceding sentence regarding the 4-fold increase in signal amplitude).

Regarding the comment that the results in cell culture “contradict the better experiment of testing Fz4 C terminal modification in vivo”, our view is that both sets of experiments are technically “correct” and it is only their interpretations that potentially contradict each other. More specifically, these experiments teach the lesson that “all that glitters (in cell culture and in vitro) is not necessarily gold (in vivo)”, a lesson that is sometimes ignored in the world of biomedical research. There are many binding interactions (like the Fz4 C-terminus and Dlg1 PDZ interaction) that are biochemically “real” but still need to be tested for their biological significance, and here is one system where that has been done. This aspect of the present body of work may recommend this paper as a good one for students to read.

2) The last experiment is a good negative result, but besides this specific non-interaction we do not learn anything concrete about Dlg1 in retinal angiogenesis or its connection to signaling pathways. Reviewer 3 encourages the authors to try new experiments to directly link Dlg1 with components of the Wnt pathway and so offer something more than speculation.

That is a valid criticism. The current study has tested the most favored model – that the PDZ-binding motif in Fz4 interacts directly with the PDZ domain of Dlg1 in vascular endothelial cells – and ruled it out as a significant mechanism for Dlg1 action in beta-catenin signaling in vivo. Now we are left with a large number of possible targets for Dlg1 action, i.e., any other intracellular or trans-membrane protein involved in beta-catenin signaling. Taking the next step to identify and validate those target(s) is going to be a major undertaking and we think that it is beyond the scope of the current manuscript.

3) Without a detailed characterisation of the expression profile of the EC from the mutant mice (which is clearly beyond the scope of this study), it is not possible to confidently conclude that the Dlg4-Fzd4 interaction does not play a role in these phenotypes. We recommend revising the interpretation of the findings to include this possibility in the interpretation of the results. Given the extensive section on possible alternatives to Fzd4 in the Discussion, it would be tempting to ask for some evidence for a role for APC or Gpr124 in this study; however, this is likely to be the focus of following papers.

That is a valid point. We agree that a Dlg1-Fz4 interaction could exist in vivo in ECs and that eliminating that interaction could produce a phenotype that is so subtle that we have missed it based on our histologic analyses. A transcriptome analysis could reveal such a phenotype, if it exists. As suggested, we have modified the Discussion to include this possibility. We agree that genetic analyses with Dlg1 and APC or with Dlg1 and Gpr124 are beyond the scope of the current manuscript.